# Zooplankton variability in the Strait of Georgia, Canada, and relationships with the marine survivals of Chinook and Coho salmon

**R. Ian Perry** [1,2]*, **Kelly Young**[2], **Moira Galbraith**[2], **Peter Chandler**[2], **Antonio Velez-Espino**[1], **Steve Baillie**[1]

**1** Pacific Biological Station, Fisheries and Oceans Canada, Nanaimo, B.C., Canada, **2** Institute of Ocean Sciences, Fisheries and Oceans Canada, Sidney, B.C., Canada

* Ian.Perry@dfo-mpo.gc.ca

**Data Availability Statement:** All relevant data and links to publicly available data are within the paper and its Supporting Information files.

## Abstract

The Strait of Georgia, Canada, has complex interactions among natural and human pressures that confound understanding of changes in this system. We report on the interannual variability in biomass of 12 zooplankton taxonomic groups in the deep (bottom depths greater than 50 m) central and northern Strait of Georgia from 1996 to 2018, and their relationships with 10 physical variables. Total zooplankton biomass was dominated (76%) by large-sized crustaceans (euphausiids, large and medium size calanoid copepods, amphipods). The annual anomaly of total zooplankton biomass was highest in the late 1990s, lowest in the mid-2000s, and generally above its climatological (1996–2010) average after 2011, although many individual groups had different patterns. Two latent trends (derived from dynamic factor analyses) described the variability of annual biomass anomalies underlying all zooplankton groups: a U-shaped trend with its minimum in the mid-2000s, and a declining trend from 2001 to 2011. Two latent trends also described the physical variables. The variability represented by these four latent trends clustered into two periods: 1996–2006, with generally declining zooplankton biomass and increasing salinities, and 2007–2018, with increasing zooplankton biomass and decreasing salinities. ARIMA modelling showed sea surface salinity at Entrance Island in the middle Strait of Georgia, the Pacific Decadal Oscillation, and the peak date of the spring phytoplankton bloom were significantly related to the two latent zooplankton trends. ARIMA models comparing zooplankton and physical variables with the marine survivals of four salmon populations which enter the Strait as juveniles (Chinook: Cowichan River, Puntledge River, Harrison River; Coho: Big Qualicum River) all included zooplankton groups consistent with known salmon prey; prominent among the physical variables were sea surface salinity and variables representing the flow from the Fraser River. These regressions explained ($_{adj}R^2$) 38 to 85% of the annual variability in marine survival rates of these salmon populations over the study time period. Although sea temperature was important in some relationships between zooplankton biomass and salmon marine survival, salinity was a more frequent and more important variable, consistent with its influence on the hydrodynamics of the Strait of Georgia system.

**Funding:** This work was funded in part by grants to RIP from the Pacific Salmon Foundation (www.psf.ca), and by funding from Fisheries and Oceans Canada. The funders had no role in study design, data collection and analysis, decision to publish, or preparation of the manuscript.

**Competing interests:** The authors have no competing interests in this work.

## Introduction

The Strait of Georgia, part of the Salish Sea in southwestern British Columbia, Canada, is a crucial marine system for the ecology, economy, and culture of this region, and beyond. Historically it has been very productive, with special and endangered populations of fish and marine mammals [1]. It is also arguably Canada's most human-dominated marine system [2, 3], with important shipping and recreational fishing activities, and is a recipient of the effluents from large and small coastal communities [4]. The complex interactions among natural and human pressures on the Strait of Georgia make it difficult to understand and manage change in this system. The use of long time series of data on pressures and ecosystem responses is one way to try to disentangle these interactions [2].

The Strait of Georgia is a large semi-enclosed estuary in southwestern British Columbia, with a surface area of about 6800 km$^2$, a volume of about 1100 km$^3$, and a maximum depth of 450 m located in its central basin [5, 6]. The major source of freshwater into the Strait is from the Fraser River, which originates in the interior of British Columbia. It is a snow-melt-dominated system, with peak flows of freshwater into the Strait of Georgia typically during June. The Strait is predominately an estuarine system, with residence times for water in the upper 50 m on the order of a few months, residence times for intermediate water (50–200 m) on the order of half a year, and residence times for the deep water of about one year [6].

Zooplankton, along with phytoplankton (e.g. [7]), form the base of this marine ecosystem and are expected to be impacted more by natural rather than direct human pressures, at least at annual to decadal time scales. The increasing availability of long time series of biological oceanographic variables (e.g. the IOC-UNESCO International Group for Marine Ecological Time Series (IGMETS) project; http://igmets.net/; [8]) is improving understanding of how zooplankton fluctuations are driven by physical variability and how, in turn, zooplankton variability impacts fish and other marine populations. In the Strait of Georgia, Chinook (*Oncorhynchus tshawytscha*) and Coho (*O. kisutch*) salmon populations have been declining since the early 1990's [9, 10], and there is much discussion about the causes of this decline (e.g. whether it is due to insufficient zooplankton prey, predation by seals, etc.; [11, 12]). The implications of these declines have significant cultural and societal importance, including concerns about endangered Southern Resident Killer Whale (*Orcinus orca*) population preferences for Chinook salmon as prey [13].

Mackas et al. [14] described the variability of the major groups of zooplankton in the Strait of Georgia from 1990 to 2010. They found this variability to be significantly related to the North Pacific Gyre Oscillation (NPGO) climate index and temperature anomalies throughout the water column. They also concluded that this zooplankton signal was related to survival anomalies of Pacific Salmon (*Oncorhynchus* spp.) and Pacific Herring (*Clupea pallasii*) stocks in the Strait of Georgia. Li et al. [15] examined a subset of these data from 1990 to 2007 and found the best single indicator of zooplankton community change within the surface 20 m was the Southern Oscillation Index (SOI), and the best local indicator was the day of the peak flow from the Fraser River. Both Mackas et al. [14] and Li et al. [15] concluded that local environmental processes were only weakly related to zooplankton variability in the Strait of Georgia, but that further research on this topic was necessary. Perry and Masson [2] identified six variables (sea surface temperature, wind speed, the North Pacific Gyre Oscillation index, human population surrounding the Strait of Georgia, recreational fishing effort, and the number of hatchery releases of Chinook salmon into the Strait) that accounted for a significant proportion of the annual patterns of variability of 37 biophysical and human data series from this ecosystem over the period 1970 to 2010. However, because of data limitations, zooplankton were not included in their analysis.

Since these studies were published, the Strait of Georgia has experienced a period of warming sea temperatures [16], driven in part by the marine heat wave of 2015–2016 in the NE Pacific [17]. In the present study we examined high-quality, full water column zooplankton taxonomic data from 1996 to 2018 to re-evaluate and update the relationships identified by Mackas et al. [14] and Li et al. [15] for the Strait of Georgia, and to evaluate the relationships of zooplankton with Chinook and Coho early marine survival patterns. Our specific objectives were (1) to identify the interannual patterns of zooplankton species composition and biomass in the Strait of Georgia over the period 1996 to 2018; (2) to determine how these patterns varied with changes in physical conditions; and (3) to explore how variations in these physical and zooplankton properties were related to the marine survival of Chinook and Coho salmon that enter into and migrate through the Strait of Georgia as juveniles. This study is a contribution to the Salish Sea Marine Survival Program led by the Pacific Salmon Foundation (Canada) and Long Live the Kings (United States).

## Methods

### Zooplankton

**Sampling.** Zooplankton sampling in the Strait of Georgia since 1996 has occurred sporadically at many locations (Fig 1). Since 2015, sampling has been conducted approximately bi-weekly to monthly from February or March to October at 17 locations in the Strait of Georgia from south of Campbell River to Juan de Fuca Strait (Fig 1). Sampling used a variety of gear types, net mesh sizes, and tow profiles. For this study, we selected vertical tow data from SCOR, Bongo, or ring nets with mouth diameters approximately 50 cm and mesh sizes approximately 250 μm. A ring net has a cone shape, whereas SCOR and Bongo nets have a cylindrical section at the top before tapering to the cod end. All nets are expected to have similar capture efficiencies [18]. Typical vertical tow profiles went from about 10 m above bottom to the surface. Nets were retrieved at 1 meter per second, and calibrated flow meters were used to record the volumes filtered by each tow. If a problem occurred with the flow meter, the volume was calculated based on net diameter, assumed filtration efficiency, and depth of tow. This occurred in less than 13% of the tows used for this analysis. All zooplankton collected were preserved immediately in 10% sodium borate-buffered formalin in seawater.

We restricted our analyses to the period 1996 to 2018 as these years had the most consistent taxonomic identifications, and data after 2018 were not fully available for this study. We also restricted our analyses to the deep (bottom depths deeper than 50 m) central and northern regions of the Strait of Georgia (Fig 1), as these had the most consistent data over this time period. In addition to this regional focus, and because many of the important zooplankton species in the Strait of Georgia undertake diel and seasonal vertical migrations, we included data from tows which sampled as much of the water column as possible. The mean water depth among our sampling locations within these deep regions was 290 m. We therefore selected vertical tows from our database which included more than half of the average water column depth, i.e. those with start depths greater than 150 m during either daytime or at night. For locations with bottom depths less than 150 m, we selected tows which covered 70% or more of the water column.

**Taxonomic analyses.** In the laboratory, samples were examined using stereomicroscopes (Wild dissecting microscope up to 2013, and Zeiss SteREO Discovery 8 thereafter) and processed in two parts based on zooplankton size. Whole samples were first scanned for large (>5 mm) or rare individuals, then split using a Folsom splitter to approximately 100 individuals greater than 5 mm. Plankton in these subsamples were identified to species, sex and stage where possible, and removed. The remaining subsample (containing organisms <5 mm) was

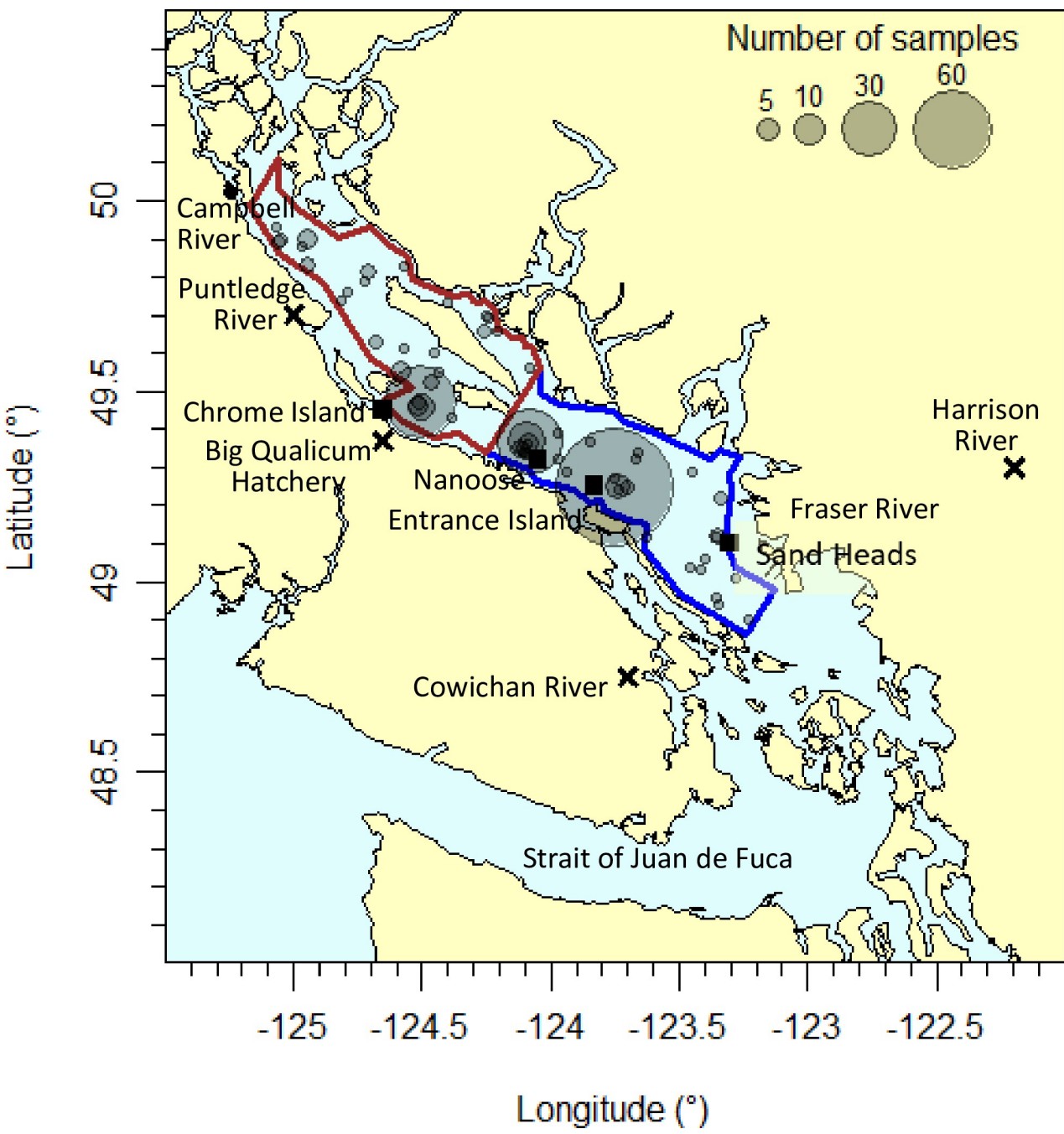

**Fig 1. Location of the study and place names.** Blue outline represents bottom depths greater than 50 m in the central Strait of Georgia, brown outline represents bottom depths greater than 50 m in the northern Strait of Georgia. Expanding symbols indicate the number of observations at each location (maximum number of observations at a location was 131). Black squares indicate the locations of the oceanographic or atmospheric data, and "X" marks locations of the salmon data used in this study. Map was created using code and bathymetric data in the R package PBSmapping [19].

split to approximately 400 individuals which were identified to the lowest possible taxonomic classification and life history stage or size class. Data were obtained from the Fisheries and Oceans Canada Institute of Ocean Sciences Zooplankton Database (version 9, data downloaded on 21 May 2020). For each taxon – stage or taxon – size class combination in each sample, biomass was calculated from the abundance and the average biomass of individuals in that

sample, derived from measured or literature values (these latter may have included the same or similar species from elsewhere). These data are available on the Government of Canada open data portal under Strait of Georgia zooplankton (https://tinyurl.com/SofGdata). We used both abundance and biomass of each taxon and stage or size, calculated as numbers or weight (dry mass in grams) per unit surface area (i.e. per $m^2$).

Zooplankton were aggregated into taxonomic groups following Mackas et al. [14]. These groups are identified in Table 1; detailed definitions are in S1 Table. Note that early life stages of large calanoid copepods (prosome lengths greater than 3 mm) were included in the small (< 1 mm) or medium (1–3 mm) calanoid copepod categories, depending on their prosome lengths [14]. Large, actively swimming adult euphausiids are often under sampled by vertical-tow zooplankton gear, in particular during daytime when the animals can see and avoid the net. We therefore multiplied our daytime adult euphausiid (adult males and females, and stages with lengths greater than 10 mm) abundances and biomass by 3 to adjust for this under sampling. This is the factor used by Mackas et al. [14] to correct for daytime net avoidance by adult euphausiids, based on a previous analysis [20]. In addition, in a much larger set of samples for the adjacent west coast of Vancouver Island (3545 samples, spanning the years 1979 to 2019 and bottom depths < 301 m), and for the same species of large euphausiids, there was a significant difference of biomass (mg m$^{-3}$) between daylight and night time periods (ANOVA on log$_{10}$-transformed biomass as mg m$^{-3}$ to normalise the variance, F = 111.1, P<<0.001). The mean (untransformed) biomass at night was 8.89 mg m$^{-3}$, and during daytime it was 3.16 mg m$^{-3}$, for a ratio night/daytime of 2.81. Since adult euphausiids in the Strait of Georgia are expected to behave similarly to those along the west coast of Vancouver Island with respect to avoidance of the same plankton gear, we applied the factor of 3 to adjust for potential daytime net avoidance by adult euphausiids in our study area.

Net avoidance by other taxa will depend on their size and swimming abilities. Calanoid copepods and amphipods are the more common larger zooplankton (along with euphausiids) in the Strait of Georgia. However, Shaw and Robinson [20] found few significant day-night differences in catchability of calanoid copepods by bongo nets in the study region, and Keen [21] concluded that net avoidance by copepods and amphipods on the north coast of British Columbia was a negligible concern for his study (although he used a plummet net with similar design to our SCOR net; plummet nets sample on the downcast). Consequently, we corrected for daytime net avoidance only for euphausiids.

**Statistical analyses.** Many species of zooplankton have strong seasonal patterns, such as multiple generations per year (e.g. *Pseudocalanus* spp.) or life stage vertical migrations (e.g. *Neocalanus* spp.). To accommodate such seasonal variability into analyses of annual patterns and interannual variability, we followed the practice of Mackas et al. [14] and other authors (e.g. [22]) and converted our zooplankton data into annual biomass anomalies. Such anomalies represent the difference of plankton biomass in a given year from the mean value of biomass over a set of years. It has the additional statistical property of not being bounded by zero as is absolute biomass. The number of locations sampled in some years was low, however. To improve the calculation of within year anomalies we aggregated data into seasonal (3 month) time periods, with winter as December (of the previous year)-January-February, spring as March-April-May, summer as June-July-August, and fall as September-October-November. To be considered as a valid year for the analyses, there had to be at least one observation in either spring or summer. We then calculated seasonal anomalies of each of the taxonomic groups over the period 1996 to 2018, and took their annual averages. This removes the seasonal signal and reduces many of the problems of irregular sampling [23]. Current practice in oceanography is to define the climatology period as being 1981–2010 (30 years). Since our data started in 1996, we used the 15 year period 1996–2010 as our climatology for calculating the

**Table 1. Zooplankton taxonomic groups and physical time series considered in the analyses.**

| Zooplankton taxonomic groups | Abbreviated name |
|---|---|
| Amphipoda—Gammarids | AmphiGam |
| **Amphipoda—Hyperiids** | **AmphiHyp** |
| Larvae of benthic animals ("Benthic larvae") | BenthicLarv |
| **Calanoid Copepods–Large (prosome length >3 mm)** | **CalCops.larg** |
| **Calanoid Copepods–Medium (prosome length 1–3 mm)** | **CalCops.med** |
| **Calanoid Copepods–Small (prosome length <1 mm)** | **CalCops.smal** |
| Cephalopoda | |
| **Chaetognatha** | **Chaetognatha** |
| **Cladocera** | **Cladocera** |
| **Ctenophora** | **Ctenophora** |
| **Euphausiidacea** | **Euphs** |
| **Fish (eggs/larval/juvenile)** | **Fish** |
| Larvacea | Larvacea |
| **Medusae** | **Medusae** |
| Mysidacea | |
| Natantia ("Shrimps") | Natantia |
| Non-Calanoid Copepods | NonCalCops |
| **Ostracoda** | **Ostracoda** |
| **Polychaeta (Pelagic)** | **PolychaetPelagic** |
| Pteropoda | Pteropods |
| Repantia ("Larval crabs") | Repantia |
| Scyphozoa | |
| Siphonophorea | Siphono |
| **Total Zooplankton biomass** | **TotBiom** |
| **Regional physical variables** | |
| **Pacific Decadal Oscillation (PDO)** | **PDO** |
| North Pacific Gyre Oscillation Index (NPGO) | NPGO |
| Multivariate ENSO Index (MEI) | MEI |
| Oceanic Nino Index (ONI) | ONI |
| Southern Oscillation Index (SOI) | SOI |
| **Local physical variables** | |
| **Vertically-averaged temperature (0–300 m near Nanoose Bay)** | **Nan.vT** |
| **Fraser River peak flow day** | **FR.pkDay** |
| **Fraser River (annual) flow volume** | **FR.flow** |
| Sand Heads air temperature | SH.airT |
| Sand Heads wind speed | SH.WSpd |
| **Sand Heads wind stress** | **SH.WStr** |
| **Entrance Island Sea Surface Temperature (SST)** | **Entr.SST** |
| **Entrance Island Sea Surface Salinity (SSS)** | **Entr.SSS** |
| **Chrome Island Sea Surface Temperature (SST)** | **Chrom.SST** |
| **Chrome Island Sea Surface Salinity (SSS)** | **Chrom.SSS** |
| **Day of year of modelled spring bloom peak** | **BloomDay** |

Variables in grey and without abbreviated names were excluded at the beginning due to too few observations (>3 years with 0 observations). Prior to statistical analyses, a second exclusion (also in grey but including abbreviated names) was made for variables which were highly (r>|0.60|) correlated with several other variables to reduce multi-collinearity issues. Abbreviated names are also indicated. Definitions of zooplankton taxa and size classes that are included in each of the taxonomic groups are provided in S1 Table.

seasonal anomalies. We also followed the updated International Council for the Exploration of the Seas (ICES) method for the calculation of the anomalies, in which the arithmetic mean biomass for a taxonomic group was calculated for each season in each year and then $\log_{10}$ transformed [23]. Because of the log transformation, we followed the ICES method to adjust for true zero (rather than missing) values, in which zeros were replaced with a random value greater than 0 and less than one-half of the lowest non-zero value in the data for that variable [23]. The equations used to calculate annual biomass anomalies were (see Table 2 for definitions of symbols):

$$\bar{Z}_{s,y} = log_{10}[\sum(Z_{s,y})/n_{s,y}] \tag{1}$$

$$\bar{\bar{Z}}_s = \sum_y(\bar{Z}_{s,y})/n_s$$

$$Z'_{s,y} = \bar{Z}_{s,y} - \bar{\bar{Z}}_s$$

$$Z'_y = \sum_s(Z'_{s,y})/n_y$$

A one-unit change in these $\log_{10}$-scaled biomass anomalies therefore represents a 10-fold change in the actual biomass.

To reduce potential problems that may be caused by many years in which a zooplankton group was not observed, we removed three taxonomic groups with sparse observations over the time period. These were (number in brackets is the number of years from 1996 to 2018 with zero occurrences): cephalopods (5), scyphozoans (15), and mysids (16). Annual anomalies of all remaining zooplankton groups are available in S2 Table. To reduce potential problems due to multi-collinearity of variables, we did a correlation analysis among all pairs of remaining zooplankton groups and removed those which were highly correlated ($r > |0.60|$) with several other groups (e.g. [24]; S3 Table). For example, cladocerans were retained because they were highly correlated with only one other group (siphonophores), whereas siphonophores were dropped because they were highly correlated with six other groups (S3 Table). This left 12 taxonomic groups in our analyses: hyperiid amphipods, large calanoid copepods, medium calanoid copepods, small calanoid copepods, chaetognaths, cladocerans, ctenophores, euphausiids, fish larvae, hydromedusae ("medusae"), ostracods, and pelagic polychaetes (Table 1).

We used dynamic factor analysis (DFA) to identify the interannual patterns (latent trends) of variability in the zooplankton biomass anomalies. Since these annual anomalies are time series, often with statistically significant autocorrelation, we used DFA with VARIMAX

**Table 2. Definitions of symbols used in Eq 1 to describe the calculations of zooplankton biomass anomalies.**

| | |
|---|---|
| $Z_{s,y}$ | Zooplankton biomass observed in season (s) and year (y) |
| $n_{s,y}$ | Number of observations of zooplankton biomass in season (s) in year (y) |
| $\bar{Z}_{s,y}$ | Log$_{10}$ transform of the arithmetic mean of zooplankton biomass in season (s) in year (y) |
| $\bar{\bar{Z}}_s$ | Mean log$_{10}$ scaled biomass in season (s) over all climatology years (y = 1996 to 2010) |
| $n_s$ | Number of years (1996 to 2010) with zooplankton biomass observations in season (s) |
| $n_y$ | Number of seasons with zooplankton biomass observations in year (y) |
| $Z'_{s,y}$ | Log$_{10}$ scaled biomass anomaly for season (s) in year (y) |
| $Z'_y$ | Mean log$_{10}$ scaled zooplankton biomass anomaly across all seasons (with observations) in year (y) |

rotation as implemented in the MARSS R package [25, 26]. This process tested from 1 to 9 combinations of latent trends and variance-covariance matrices (equal, diagonal and equal, diagonal and unequal, or unconstrained). The Akaike weights and corresponding changes in the AICc (Akaike's Information Criterion with a correction for small sample sizes) were used to identify the combinations of trends and variance-covariance matrices up to a cumulative Akaike weight of 1. We selected as the best model the number of factors and the variance–covariance matrix with the lowest delta AICc and Akaike weight. Version 3.6.0 of the statistical programming language R [27] was used for all analyses.

## Physical data

**Temperature and salinity.** Sea surface temperature and salinity data for the period 1996 to 2018 were obtained from measurements taken daily at one lighthouse station in each of the central and northern Strait of Georgia (Entrance Island, Chrome Island, Fig 1; https://www. dfo-mpo.gc.ca/science/data-donnees/lightstations-phares/index-eng.html). In addition, to represent temperature throughout the water column, we used vertically-averaged temperature obtained from approximately bi-weekly near-bottom to surface CTD casts conducted by the Canadian military in deep water (approximately 300 m bottom depth) in the middle of the Strait of Georgia near Nanoose (Fig 1).

**Atmospheric data.** Hourly air temperature and wind speed for the period 1996 to 2018 were obtained from the Environment and Climate Change Canada weather station at Sand Heads near Vancouver International Airport at the mouth of the Fraser River (Fig 1; http:// climate.weather.gc.ca/historical_data/search_historic_data_e.html). Wind speed was converted to wind stress using

$$\tau_{wind} = \rho_{air} * C_D * U^2 \tag{2}$$

in which $\tau_{wind}$ is the wind stress in Newtons m$^{-2}$, $\rho_{air}$ is the density of air (1.22 kg m$^{-3}$), $C_D$ is a dimensionless wind-drag coefficient (0.0013), and $U$ is the wind speed (m s$^{-1}$) (e.g. [28]).

**Fraser River flow.** The Fraser River is the largest source of fresh water into the Strait of Georgia, and has a dominating effect on the circulation and dynamics of the southern half of the Strait. We used two measures of fresh water flow from the Fraser River: the monthly mean flow volumes as measured near Hope, B.C., at the head of the Fraser Valley (Station 08HF005: 49.38˚N 121.45˚W; https://wateroffice.ec.gc.ca/search/historical_e.html), and the day of the peak flow each year (calculated from the daily flow volumes). This second measure was used to represent changes in the timing of the flow into the Strait of Georgia each year.

**Date of the spring phytoplankton bloom.** The phytoplankton bloom in spring is the beginning of the productive season in temperate ecosystems. In the Strait of Georgia there is evidence that variations in the timing of the spring phytoplankton bloom can impact the survival and production of young of the year Pacific Herring [29]. We used the mean modelled date for the peak of the spring phytoplankton bloom (hereafter called the date of the spring bloom) from 1996 to 2018 from Allen et al. [30]. These estimates are based on a one-dimensional vertical mixing model coupled to a nitrate-diatom biological model which is tuned for a deep-water location in the central Strait of Georgia [31].

**Climate indices.** Several studies [2, 14, 15] have found statistically significant relationships between various climate indices for the North Pacific and local processes in the Strait of Georgia. We included several of these climate indices (all data except for the NPGO can be accessed via https://www.esrl.noaa.gov/psd/data/climateindices/list/):

- Pacific Decadal Oscillation (PDO) represents the first empirical orthogonal function of monthly sea surface temperature anomalies in the North Pacific, and tracks east-west warm-cool ocean conditions in the region [32];

- Oceanic Nino Index (ONI), the Multivariate ENSO Index (MEI), and the Southern Oscillation Index (SOI) represent different manifestations of El Niño–Southern Oscillation (ENSO) dynamics in the tropical Pacific Ocean. The ONI represents a 3-month running mean of the average sea surface temperature (SST) anomalies in the east-central equatorial Pacific (the Niño 3.4 region). The MEI combines meteorological and oceanographic data to characterise the intensity of an ENSO event in the tropics. The SOI is calculated from the sea level pressure difference between Tahiti and Darwin, and also indicates the intensity of El Niño events. For both the ONI and MEI, large positive values represent El Niño conditions, and large negative values represent La Niña conditions, whereas the direction is the reverse for the SOI;

- North Pacific Gyre Oscillation (NPGO) represents the second empirical orthogonal function of sea surface height in the NE Pacific. Its variation reflects changes in the North Pacific gyre circulation and therefore changes in large-scale wind-driven upwelling and horizontal advection [33]. NPGO data are available at http://www.o3d.org/npgo/.

All these physical data were reduced to seasonal averages, using the same definition for season as applied to the zooplankton data. Normalised seasonal anomalies were calculated using a base period of 1996 to 2010 by:

$$Vanom_{season,year} = [V_{season,year} - \bar{V}_{season}]/sd(V_{season}) \tag{3}$$

in which $Vanom_{season,year}$ is the anomaly for a physical variable in a season and year, $V_{season,year}$ the value of that variable in a season and year, $\bar{V}_{season}$ is the mean value of that variable in a season over all years in the base (climatology) period, and $sd(V_{season})$ is the standard deviation of that variable for a season over all years in the base period.

Annual anomalies were then calculated as the mean of the anomalies for that variable across seasons within each year (S2 Table). Annual anomalies of all physical variables were screened for multi-collinearity and highly correlated (r > |0.60|) variables (S3 Table) were removed from the statistical analyses, in the same manner as for the zooplankton data. The physical time series we considered in our analysis are listed in Table 1. These data were also analysed using dynamic factor analysis, as described above, to identify significant latent (underlying) trends.

## Salmon marine survivals

The focal salmon species for the Salish Sea Marine Survival Program are Chinook and Coho, because of their strong declines in the 1990's and failures to recover (e.g. [4]). For Chinook, we used the stock-specific early marine survival data, defined as the proportion of the population that has survived from release from the hatchery to Age 2. We used ocean entry years (calculated as brood year + 1) 1996 to 2009 presented by Ruff et al. [9] updated to ocean entry year 2015 with data from the Pacific Salmon Commission Chinook Technical Committee [34]. These survivals were derived from coded-wire tag release and recovery data using backwards cohort reconstruction [9]. Since each Chinook brood year returns to its natal stream over a period of from one to five years, the most recent year for which complete returns are available, and therefore for which final early marine survival estimates can be made, is 2014. We present preliminary early marine survival estimates for 2015 to 2018, but these are considered highly uncertain and so were not used in our model fitting procedures. We selected stocks representative of three clusters that enter the Strait of Georgia ([9], their Fig 3): Cowichan River Fall

spawners, Harrison River Fall spawners, and Puntledge River Summer spawners. These three populations have summer or fall run timing, with a sub-yearling release strategy. With Coho, marine survival estimates are based on coded-wire tagged cohorts, and are estimated by the number of coded-wire tagged fish caught in the various marine fisheries plus the number of tagged fish in the escapement, all divided by the number of smolts released [10]. Marine survivals for Coho are therefore total marine survivals. Coho generally return the year after ocean entry, and final marine survival estimates are available a year later. We used marine survival data from the Big Qualicum River hatchery for ocean entry years from 1996 to 2017. Ocean entry year was defined as the smolt year (brood year+2).

Since these marine survival data are proportions, and therefore are bounded by 0 and 1, they were transformed to stabilise the variance prior to statistical analyses [35]. We used the logit transform:

$$\alpha = logit(p) = -logit\left(\frac{1}{p} - 1\right) \tag{4A}$$

in which p is the early marine survival rate. The back-transformation (inverse logit) of this equation is

$$logit^{-1}(\alpha) = \frac{e^{\alpha}}{e^{\alpha} + 1} \tag{4B}$$

Note that the logit transform ($\alpha$) has an inverse relationship to the instantaneous mortality rate k, where k is defined as–$\log_e$ (survival rate).

The logit-transformed data ($\alpha$) were then centred and scaled (by subtracting the mean and dividing by the standard deviation) in preparation for the regression analyses with the zooplankton and physical variables as the predictors.

## Statistical comparisons among zooplankton, physical, and salmon survival data

The latent trends calculated from the 12 zooplankton taxonomic groups and the 10 physical variables, collectively, represented underlying physical and lower trophic level patterns in the Strait of Georgia marine ecosystem. We used cluster analysis on these zooplankton and physical latent trends to determine if there were sequences of years with similar characteristics. The method for this analysis was constrained (chronological) hierarchical incremental sum of squares clustering, which kept years in their correct sequence and found significant breaks between years based, in our case, on the underlying zooplankton composition and physical factors. We used the R package 'rioja' [36] for this constrained clustering, with a Euclidean distance matrix. To determine the number of significant clusters, we used a broken-stick model in which the proportion of the variance accounted for by progressive separation of the real data was compared with the proportion predicted by groupings of years of random length [37].

Statistical comparisons among the zooplankton, physical, and salmon marine survival data were based on linear regression methods. The following approach was used:

1. tested for trend and level stationarity of the response variable using the Kwiatkowski-Phillips-Schmidt-Shin (KPSS) test. Regression statistics of the type applied in this study require the time series to be stationary around a mean or linear trend, i.e. that the mean and variance are constant over time. If either of these were significantly different from zero (P<0.05) we first-differenced all variables, and retested for stationarity. We accepted a small amount of potential non-stationarity (i.e. P values close to 0.05) because stationarity

and autocorrelation were also checked during the subsequent autoregressive integrated moving average (ARIMA) modelling, below;

2. calculated linear regressions for all possible combinations of explanatory variables and identified the top (quantified using the residual sum of squares) models with 1, 2, 3, 4, and 5 variables. We used the best subsets regression method from the R package 'leaps' [38] for this step. We did not consider interaction terms among the variables;

3. evaluated each of these top candidate models using ARIMA methods. Each of these five models (with 1, 2, 3, 4, or 5 co-variates) was analysed to identify the best model among a variety of autoregression and moving average possibilities. In this analysis, 'best' was determined by minimising the model Akaike information criterion with a correction for small sample sizes (AICc) and Maximum Likelihood Estimation (MLE) [39]. For example, for ARIMA order (p,d,q), "p" indicated whether autocorrelation was important (p>0), "d" indicated whether differencing of successive values was required (d>0), and "q" indicated whether a moving average term was needed (q>0). An ARIMA order of (0,0,0) indicated that no corrections for autocorrelation, differencing, or moving average were required. Each model was then run using its best ARIMA order to calculate the model parameters. We used the 'auto.arima' command from the R package 'forecast' [40] to run these ARIMA analyses;

4. determined the overall top model by comparing the ARIMA results among the five models (with 1, 2, 3, 4, or 5 co-variates), using the following criteria: a) since we were not concerned with the use of these models for forecasting, a valid model had to have all of its co-variates significantly different from 0 (i.e. the 95% confidence interval about the parameter estimate could not include 0); b) a low model AICc value; and c) a high P value of a Ljung-Box test for autocorrelation of the model residuals (P values greater than 0.05 preferred, i.e. showing no autocorrelation among the residuals);

5. if the order of the overall top selected model was (0,0,0), we ran a final check using OLS linear regression to determine the adjusted $R^2$ and P values. In addition, model diagnostics (residuals versus fitted values to assess the constancy of the variance, QQ plot to assess normality of the residuals, and the residuals versus leverage to assess the importance of possible outlier values) were examined. If two models appeared to be equally good models, they were compared by analysis of variance (a significant ANOVA value meant the model with more covariates was the better fit).

Since the salmon early marine survival data series were short (20 or fewer years), we used all available data to build the regression models following the above procedures, and evaluated the predictive performance of the models with 5 repeat 5-fold cross validation (using the R package 'caret'; [41, 42]). Model results were back-calculated to their original units, and plotted against the original early marine survival time series for comparison. Early marine survivals for any missing years, and for years with incomplete returns, between 1996 and 2018 were projected using the regression models and the values of the covariates for the missing years, and their 95% prediction intervals were calculated.

## Results

### Study area

Sample locations in the central and northern regions of the Strait of Georgia have varied over the years (Fig 1), depending on the objectives of the program doing the sampling. Ninety percent of the bottom depths at these locations were between 135 and 405 m (5[th] and 95[th]

**Table 3. Number of zooplankton samples for the deep central and northern regions of the Strait of Georgia by season, available within the selection parameters (see Methods).**

| Year | Winter | Spring | Summer | Fall | Total |
|---|---|---|---|---|---|
| 1996 | 2 | 2 | 0 | 0 | **4** |
| 1997 | 1 | 2 | 4 | 4 | **11** |
| 1998 | 9 | 7 | 4 | 2 | **22** |
| 1999 | 0 | 2 | 5 | 0 | **7** |
| 2000 | 0 | 2 | 2 | 0 | **4** |
| 2001 | 0 | 2 | 6 | 0 | **8** |
| 2002 | 8 | 1 | 0 | 0 | **9** |
| 2003 | 0 | 1 | 1 | 6 | **8** |
| 2004 | 2 | 14 | 3 | 8 | **27** |
| 2005 | 3 | 4 | 1 | 1 | **9** |
| 2006 | 0 | 5 | 0 | 4 | **9** |
| 2007 | 2 | 3 | 1 | 1 | **7** |
| 2008 | 2 | 4 | 11 | 13 | **30** |
| 2009 | 9 | 6 | 15 | 16 | **46** |
| 2010 | 7 | 0 | 9 | 5 | **21** |
| 2011 | 4 | 0 | 3 | 2 | **9** |
| 2012 | 0 | 3 | 8 | 6 | **17** |
| 2013 | 0 | 13 | 7 | 7 | **27** |
| 2014 | 0 | 12 | 15 | 10 | **37** |
| 2015 | 3 | 65 | 42 | 27 | **137** |
| 2016 | 10 | 59 | 57 | 30 | **156** |
| 2017 | 14 | 37 | 31 | 12 | **94** |
| 2018 | 7 | 31 | 14 | 9 | **61** |
| **Total** | **83** | **275** | **239** | **163** | **760** |

Seasons were defined as Winter (Dec of the year before, Jan, Feb), Spring (Mar, Apr, May), Summer (Jun, Jul, Aug) and Fall (Sep, Oct, Nov).

percentiles of the distribution of bottom depths), with 90% of the maximum tow depths between 123 and 400 m. The number of samples (within the selection parameters for this study, see Methods) generally exceeded 30 per year after 2013, but was much less between 1996 and 2007 (Table 3). Spring was the season with the most coverage, but there were two years with no samples in spring and eleven years with three or less samples in spring (Table 3).

## Zooplankton

Absolute abundance and biomass patterns for total zooplankton showed the highest mean values in 1999, and the lowest mean values between 2002 (for abundance) and 2005 (for biomass; Fig 2). The widest ranges of abundance and biomass among samples occurred in the years since 2014, as might be expected from the greater sampling effort spread throughout the year. Note, however, that for earlier years the annual mean abundance and biomass values (not $log_{10}$ transformed) in Fig 2A and 2B are affected by the uneven distribution of samples among seasons (Table 3). The annual biomass anomalies are therefore a better representation of annual biomass patterns because they take account of seasonal differences in sampling. These annual biomass anomalies show that the highest total zooplankton biomass was in 1998, which then declined to a minimum in 2005, and recovered to mostly positive anomalies since 2011 (Fig 2D). Seasonally, the median of total biomass was highest in summer (Fig 2C).

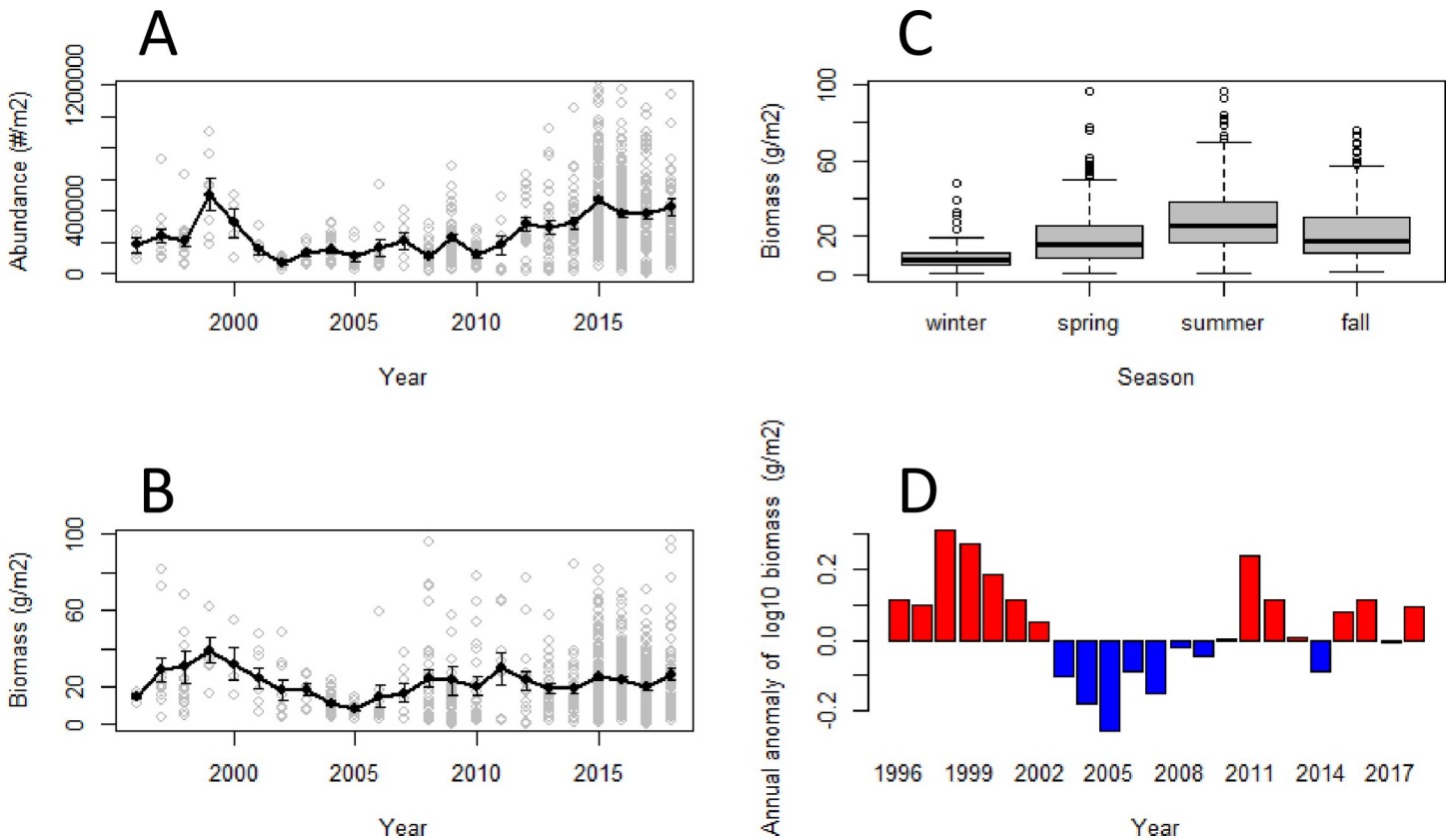

**Fig 2. Total zooplankton abundance, biomass, distribution of seasonal biomass, and annual biomass anomaly.** A) Grey symbols: abundance (number m⁻²) of all selected samples in each year. Black dots and line represent the annual mean derived from the selected samples each year, and the vertical lines represent 1 standard error about these mean annual estimates. High outlier points (representing 0.5% of the selected data) have been removed to more clearly show the spread along the Y-axis; B) Grey symbols: biomass (g m⁻²) of all selected samples in each year. Black dots and line represent the annual mean derived from the selected samples each year, and the vertical lines represent 1 standard error about these mean annual estimates. High outlier points (representing 0.5% of the selected data) have been removed to more clearly show the spread along the Y-axis; C) box and whisker plots of total biomass (g m⁻²) by season; D) Anomalies of $\log_{10}$ annual total biomass (based on the annual average of seasonal anomalies). Data range is from 1996 to 2018, with the climatology period for the anomaly calculations from 1996 to 2010.

It is possible that the uneven sampling effort among years may have influenced the interannual pattern of total zooplankton biomass anomalies. There are two questions: (1) Could the larger number of samples from 2015 to 2018 (average of 112 samples per year) have influenced the biomass anomalies compared, for example, with the period from 2003 to 2007 when only 12 samples on average were collected each year? (2) What is the probability that the sequence of low total zooplankton biomass anomalies from 2003 to 2007 (5 years in a row with anomalies less than -0.10, Fig 2D) would have occurred if the true distribution of samples was similar to that in 2015 to 2018 (i.e. with true annual anomalies near or somewhat above the climatological mean, 1996–2010) and 12 samples had been collected each year? These questions were assessed using a numerical simulation with 1000 repetitions (S1 File). The simulations demonstrate that, for question 1, the anomalies calculated using the full data set for 2015 to 2018 were almost identical to the mean anomalies that would likely have been observed if only 12 samples had been collected each year. It is therefore unlikely that the higher anomalies in 2015 to 2018 resulted only from the larger number of samples in those years, compared with the number of samples collected from 2003 to 2007. For question 2, the probability of getting five biomass anomalies in a row with values less than -0.10, as was observed from 2003 to 2007 (Fig 2D), if the true distribution of total zooplankton biomass was as observed in 2015 to 2018, and based

on 12 samples per year, is unlikely (P = 0.00032; S1 File). We conclude that the total zooplankton biomass, and therefore the biomass anomalies, in 2003 to 2007 was lower than the biomass observed in 2015 to 2018, even though an average of only 12 samples were collected each year.

On average over all years (Fig 3), euphausiids comprised the largest percentage of the total zooplankton biomass (26% as g m$^{-2}$, which included the expansion for daytime net avoidance), medium sized calanoid copepods comprised the second largest percentage of biomass (21%), and large calanoid copepods and amphipods (gammarids and hyperiids) comprised the third largest percentage of total zooplankton biomass (14% each). Euphausiid biomass was especially high in 1998, 2002, and from 2008 to 2010 (Fig 3B), when it comprised on average 56% of the total biomass in each year. Together, euphausiids, large and medium calanoid copepods, and amphipods constituted 76% on average (range 61–88%) of the total zooplankton biomass each year from 1996 to 2018. The annual anomaly patterns for other (either smaller sized or less abundant taxonomic groups) were at times very different from the pattern of total biomass, in particular with high biomass anomalies since 2011 (S1 Fig).

Changes in dominant species (as biomass, g m$^{-2}$) among years were observed within some of these key taxonomic groups. Among the large calanoid copepods, *Neocalanus plumchrus* comprised the dominant biomass in 1996–2001 and in 2011–2013; their low biomass around 2005 is consistent with their observed near-disappearance from the Strait of Georgia at that time [43]. However, *Eucalanus bungii* biomass increased in this group in 2007–2009 and in 2016–2018 (Fig 4; [44]). Among the medium size calanoid copepods, *Metridia* spp. (mostly *M. pacifica*) had the highest percent biomass from 2003–2009, but after 2011 *Metridia* spp. and *Calanus* spp. (*C. pacificus* and *C. marshallae*) had similar relative biomass contributions (Fig 4). Euphausiid biomass was largely *Euphausia pacifica* in all years; *Thysanoessa longipes* also occurred during cooler years (mostly before 2010) and *T. spinifera* in warmer years (mostly after 2009). Gammarid amphipod biomass was dominated by *Cyphocaris challengeri*, and hyperiid amphipod biomass was mostly *Themisto pacifica*. The biomass of chaetognaths was dominated by *Parasagitta elegans*, with somewhat greater diversity of species in warmer years. Pelagic polychaete biomass was mostly *Tomopteris septentrionalis*. The fish larvae group was comprised of Clupeiformes, Gadiformes, Perciformes, Osmeriformes, and Scorpaeniformes.

## Physical variables

Annual anomalies of the 10 biophysical variables that were not highly correlated with several other physical variables (Table 1) had a variety of patterns over the study period (Fig 5). These patterns ranged from highly variable (e.g. Fraser River flow, Fig 5G), to a dome-shaped pattern with highest values in the early-mid 2000s (e.g. Entrance Island sea surface salinity, Fig 5C), to periodic about an increasing (e.g. Chrome Island sea surface temperature, Fig 5B) or decreasing trend (e.g. the day of peak flow from the Fraser River, in which decreasing means earlier in the year, Fig 5H). Of particular note was that the North Pacific Gyre Oscillation anomaly (NPGO) was excluded from subsequent statistical analyses because it was highly negatively correlated with the PDO, and highly positively correlated (r = 0.64; S3 Table) with the day of the peak flow from the Fraser River. Higher positive values of the NPGO resulted in later dates for the peak flow from the Fraser River (linear regression, $_{adj}R^2$ = 0.35, P≪0.01).

## Statistical modelling of physical and zooplankton variables

Dynamic factor analysis of the 12 uncorrelated taxonomic groups was used to identify latent (underlying) trends in the zooplankton data. The best (lowest AICc) model had two latent trends and a diagonal and equal variance–covariance matrix (Table 4A). These two latent trends are shown in Fig 6. Zooplankton trend 1 presented a U-shaped pattern with a minimum

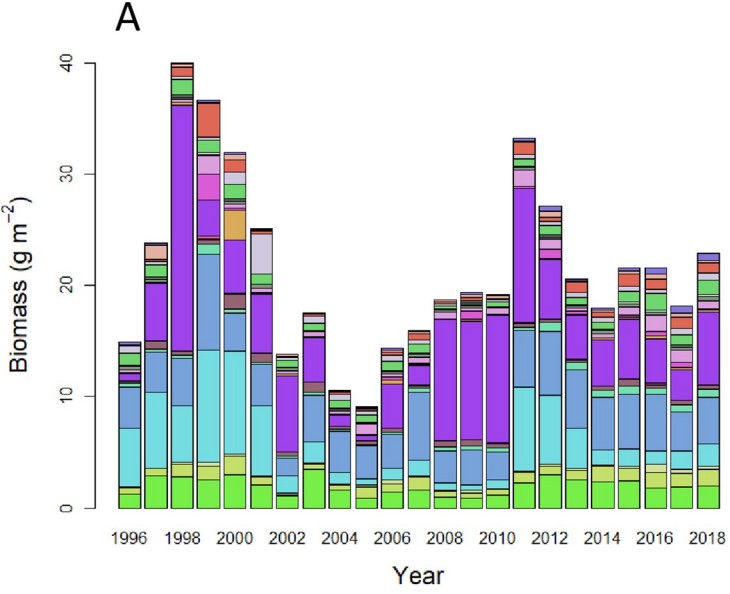

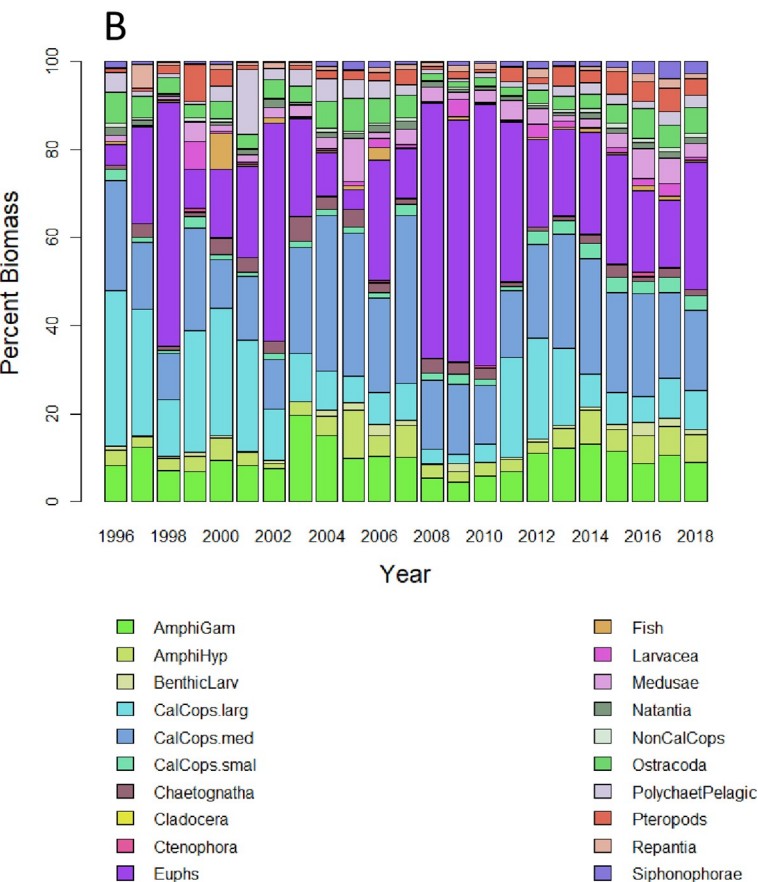

**Fig 3. Total zooplankton biomass (g m⁻²) by taxonomic group.** A) Total biomass of each group; B) percent biomass of each group. Abbreviations of taxonomic group names are defined in Table 1.

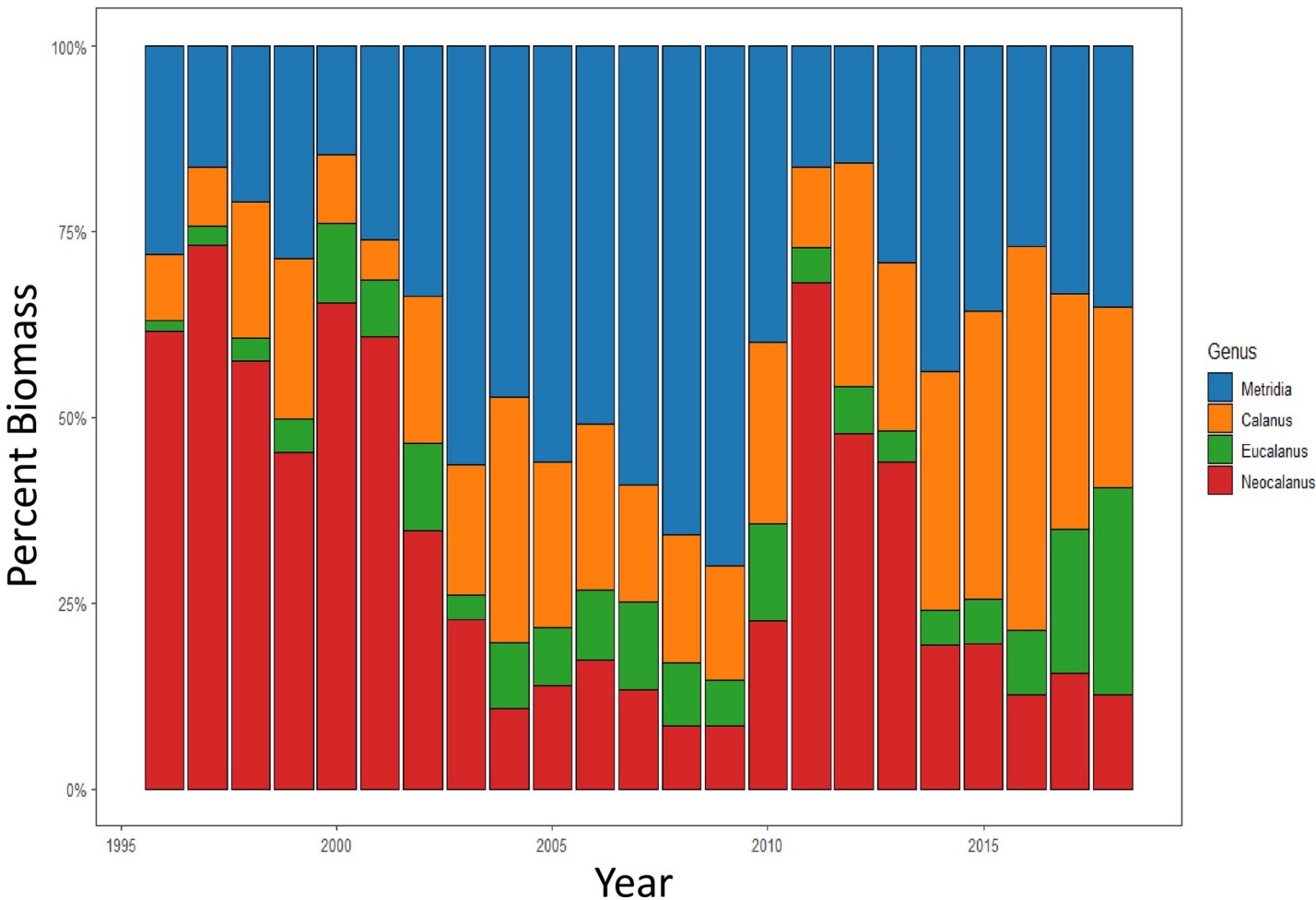

**Fig 4. Percent biomass (g m⁻²) by year for the two main genera of large and medium calanoid copepods.** Large calanoid copepod taxonomic groups: *Neocalanus* spp., *Eucalanus* spp. Medium calanoid copepod taxonomic groups: *Metridia* spp., *Calanus* spp.

in 2009 (Fig 6A). Zooplankton trend 2 presented a peak around 2001–2003 then a decline to a minimum in 2016 (Fig 6B). Zooplankton groups loading strongly on trend 1 included large and medium sized calanoid copepods, and ostracods (Fig 7A). These groups had distinct U-shaped patterns in their annual biomass anomalies (along with their strong correlates such as gammarid amphipods and non-calanoid copepods), as shown by S1 Fig. The absence of any taxonomic group with a strong negative loading on trend 1 is consistent with no group having a distinct dome-shaped pattern of annual anomalies with a peak in the mid 2000s (S1 Fig). Chaetognaths loaded positive on zooplankton trend 2 (Fig 7A), consistent with the overall decline of its annual anomalies (S1 Fig). In contrast, fish larvae loaded negative on trend 2, which is consistent with the general increase of their annual anomalies (S1 Fig). Despite being a major contributor to the total zooplankton biomass, the pattern of annual anomalies of euphausiids did not load strongly on zooplankton trends 1 or 2 (Fig 7A).

Dynamic factor analysis of the 10 physical variables found the best (lowest AICc) model also had two latent trends and a diagonal and equal variance–covariance matrix (Table 4B). Physical trend 1 presented a sinusoidal-like pattern, with a periodicity of about 10 years (Fig 6C). Physical trend 2 peaked in 2001 and then slowly declined until flattening out after 2011 (Fig 6D). Physical variables that loaded strongly positive on trend 1 included sea surface

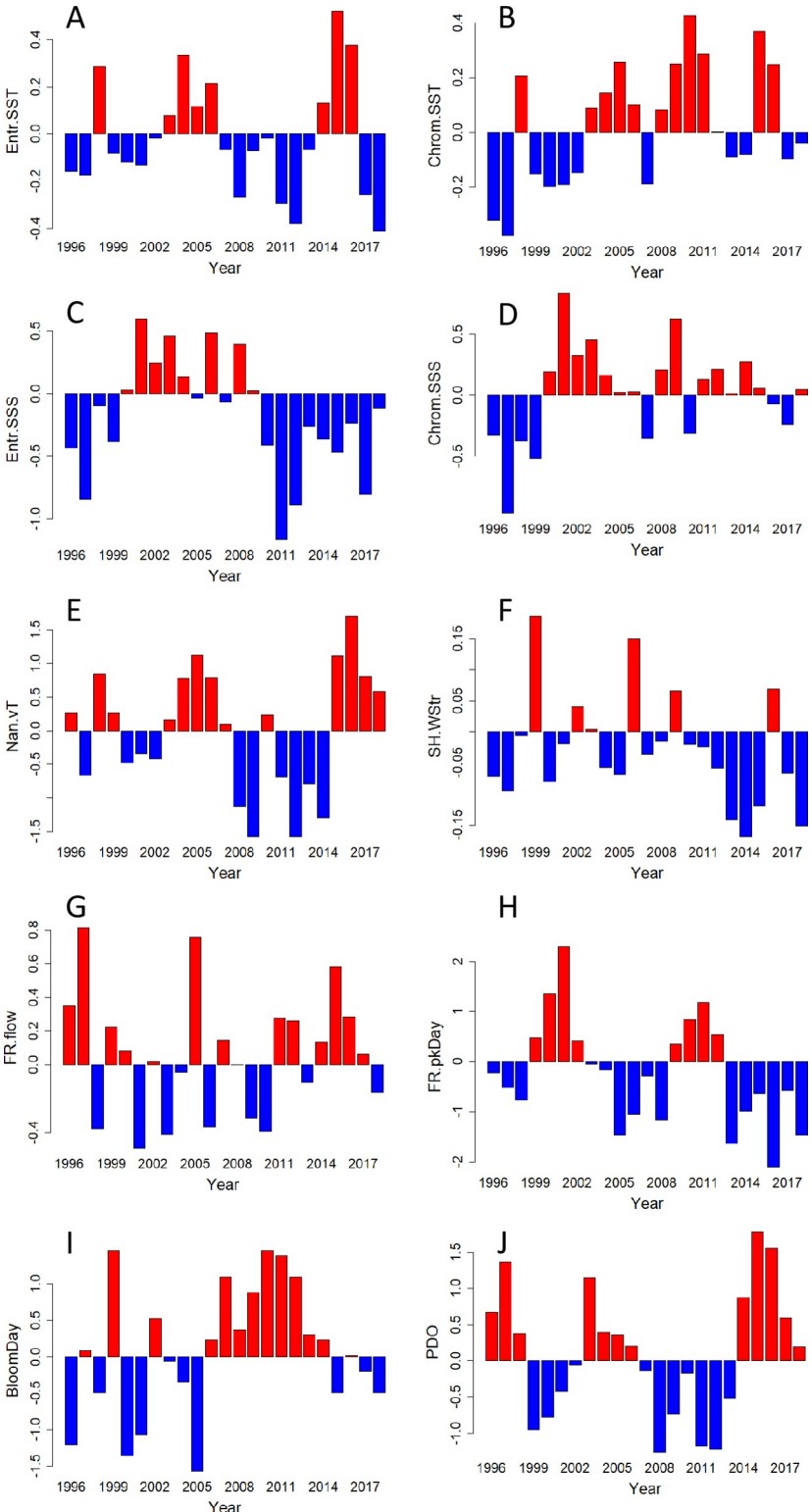

**Fig 5. Annual anomalies of physical variables, 1996–2018.** A) sea surface temperature (SST) at Entrance Island, b) sea surface temperature (SST) at Chrome Island, c) sea surface salinity (SSS) at Entrance Island, D) sea surface salinity (SSS) at Chrome Island, E) vertically-averaged temperature measured off of Nanoose, F) wind stress measured at Sand Heads light station at the mouth of the Fraser River, G) annual fresh water flow volume from the Fraser River (measured at Hope), H) day of the peak flow from the Fraser River (measured at Hope), I) modelled date for the peak

of the spring phytoplankton bloom in the central Strait of Georgia, J) Pacific Decadal Oscillation index. Abbreviations of physical variable names (Y-axis labels) are defined in Table 1.

temperature at Entrance Island, vertically-averaged sea temperature measured near Nanoose Bay, and the Pacific Decadal Oscillation. The day of the peak flow in the Fraser River and the (modelled) timing for the spring phytoplankton bloom both loaded strongly negative on physical trend 1 (Fig 7B). In contrast, sea surface salinities measured at Entrance and Chrome Islands loaded strongly positive on physical trend 2, whereas the total annual fresh water flow from the Fraser River loaded strongly negative on physical trend 2 (Fig 7B). This is consistent with decreasing fresh water flowing from the Fraser River producing increased sea surface salinities in the Strait of Georgia, and vice versa. Taken together, these results suggest that physical trend 1 represented a sea temperature and event timing (spring bloom peak, day of peak flow in the Fraser River) axis, and trend 2 represented a sea surface salinity and vertical stratification axis.

The results of the constrained (chronological) cluster analysis based on the two latent zooplankton trends and the two latent physical trends revealed a maximum of four significant clusters (Fig 8A; note that the largest number of significant clusters was determined as the highest number of groups whose sum of squares exceeded the values obtained from a randomisation process). The strongest evidence was for two clusters at a Euclidean distance greater than 40, which split the time series into the years 1996–2006, and 2007–2018 (Fig 8B). The

**Table 4. Dynamic factor analyses of the 12 zooplankton taxonomic groups (A), and of the 10 physical variables (B).** Each combination of variance-covariance matrix configurations, and number of latent (underlying) trends, was ranked according to their best fit statistics (log-likelihood and Akaike Information Criterion with the correction for small sample sizes (AICc)). Only the top 10 of 44 trials are shown for (A), and the top 10 of 36 trials for (B). The top-ranked model is in the first row in each table.

| A | | | | | |
|---|---|---|---|---|---|
| Variance—Covariance matrix | Number of trends | log Likelihood | delta AICc | Akaike weight | Cumulative Akaike weight |
| diagonal and equal | 2 | -338.83 | 0.00 | 6.07E-01 | 0.607 |
| diagonal and unequal | 2 | -326.08 | 2.21 | 2.01E-01 | 0.808 |
| equalvarcov | 2 | -338.83 | 2.42 | 1.81E-01 | 0.990 |
| diagonal and equal | 3 | -331.06 | 9.56 | 5.10E-03 | 0.995 |
| equalvarcov | 1 | -356.00 | 11.17 | 2.28E-03 | 0.997 |
| diagonal and unequal | 3 | -317.30 | 12.15 | 1.39E-03 | 0.998 |
| equalvarcov | 3 | -331.06 | 12.18 | 1.38E-03 | 1.000 |
| diagonal and equal | 4 | -322.41 | 16.69 | 1.44E-04 | 1.000 |
| equalvarcov | 4 | -322.26 | 19.23 | 4.06E-05 | 1.000 |
| diagonal and equal | 1 | -361.22 | 19.39 | 3.74E-05 | 1.000 |
| B | | | | | |
| Variance—Covariance matrix | Number of trends | log Likelihood | delta AICc | Akaike weight | Cumulative Akaike weight |
| diagonal and equal | 2 | -295.91 | 0.00 | 2.75E-01 | 0.275 |
| diagonal and equal | 1 | -306.48 | 0.33 | 2.34E-01 | 0.509 |
| equalvarcov | 2 | -295.38 | 1.36 | 1.39E-01 | 0.649 |
| diagonal and equal | 3 | -286.75 | 1.73 | 1.16E-01 | 0.764 |
| equalvarcov | 3 | -285.61 | 2.07 | 9.76E-02 | 0.862 |
| diagonal and unequal | 2 | -286.27 | 3.40 | 5.03E-02 | 0.912 |
| diagonal and equal | 4 | -278.42 | 3.99 | 3.74E-02 | 0.950 |
| diagonal and unequal | 3 | -275.91 | 4.62 | 2.73E-02 | 0.977 |
| equalvarcov | 4 | -277.80 | 5.55 | 1.71E-02 | 0.994 |
| diagonal and unequal | 1 | -299.79 | 7.76 | 5.70E-03 | 1.000 |

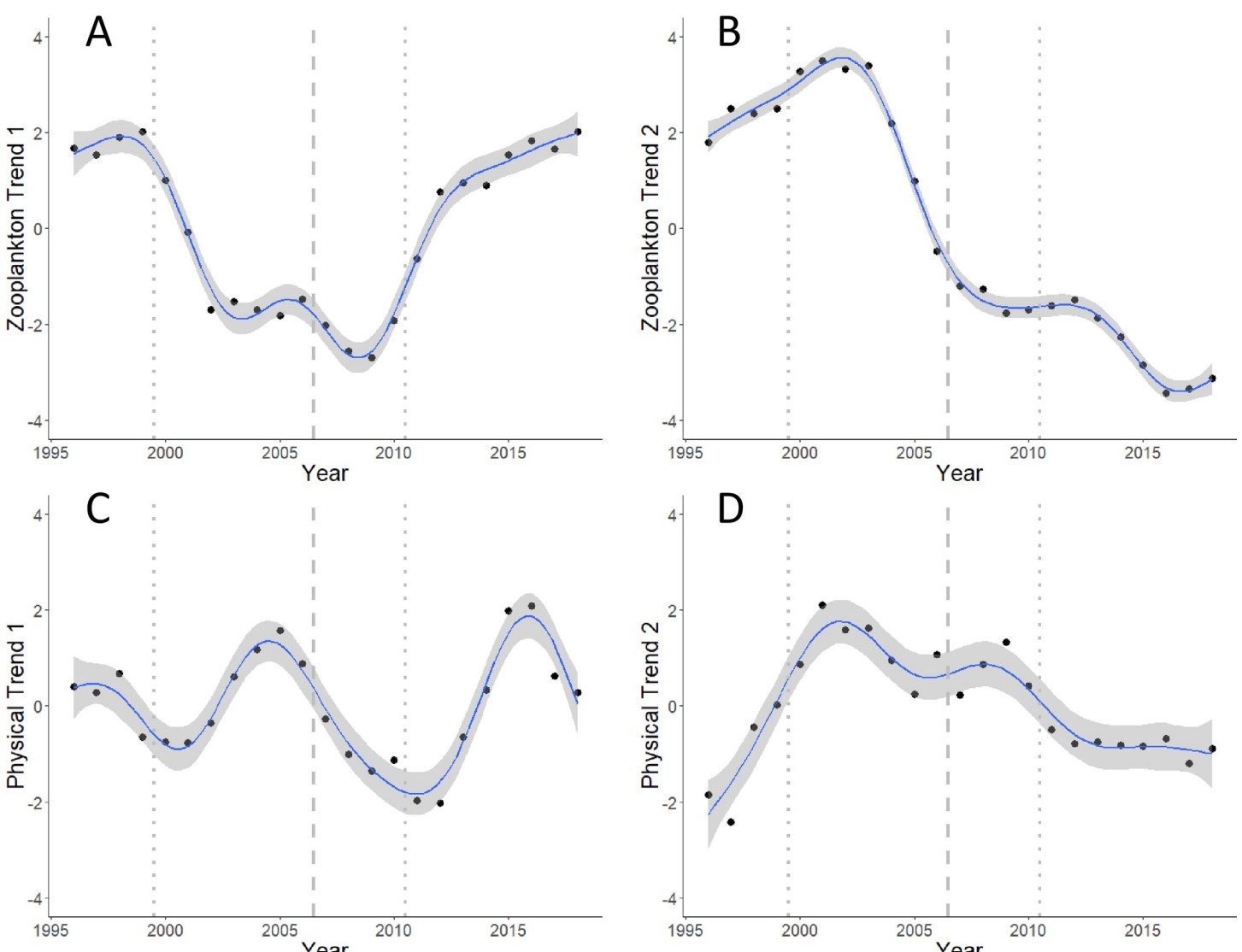

**Fig 6. Latent (underlying) trends derived from the 12 zooplankton taxonomic groups, and the 10 physical variables.** (A, B) Zooplankton groups; (C, D) physical variables. Black dots represent the trends derived for each year; blue line and shading represent 95% confidence bands about a loess smoother applied to these annual values (derived from a general additive model with year as the independent variable) to better show the general patterns. Dashed line between 2006 and 2007 represents the major two-cluster divisions identified in the chronological cluster analysis, and the dotted lines represent the other significant separations between groups of years in the four-cluster model (see Fig 8, below).

four cluster model grouped years into the periods 1996–1999, 2000–2006, 2007–2010, and 2011–2018 (Fig 8B). The breaks between clusters imply that significant changes occurred in the trends underlying the environmental conditions of the deep water areas of the Strait of Georgia, as represented by these 12 zooplankton groups and 10 physical variables. These breaks are shown by the vertical lines on the trends for the zooplankton and physical variables in Fig 6. The major break between 2006 and 2007, shown by the dashed lines (Fig 6), separated distinct patterns of variability in these trends (e.g. separated the decreasing and increasing legs of zooplankton trend 1, the inflection point of zooplankton trend 2, and one period of the cyclical pattern of physical trend 1).

Using the approach described in Methods, we found no significant regression relationships (P greater than 0.05) between the two zooplankton latent trends and the 10 physical variables,

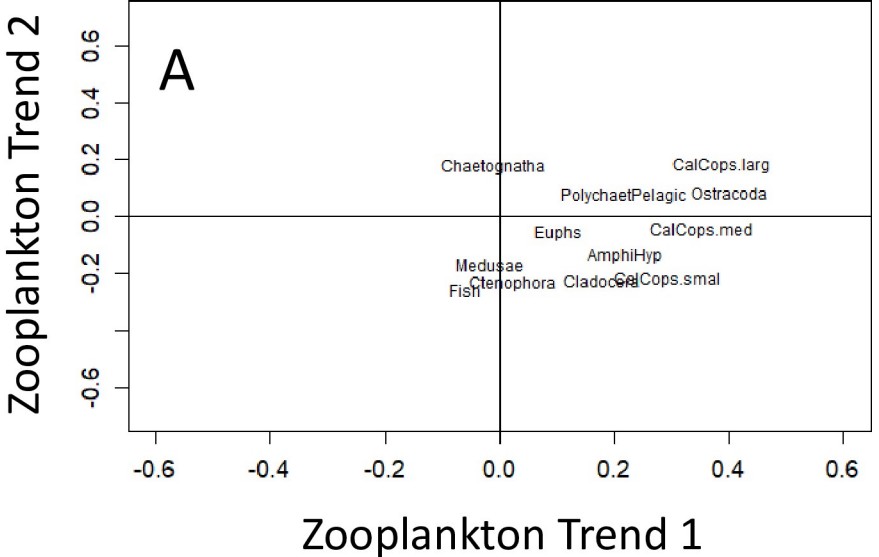

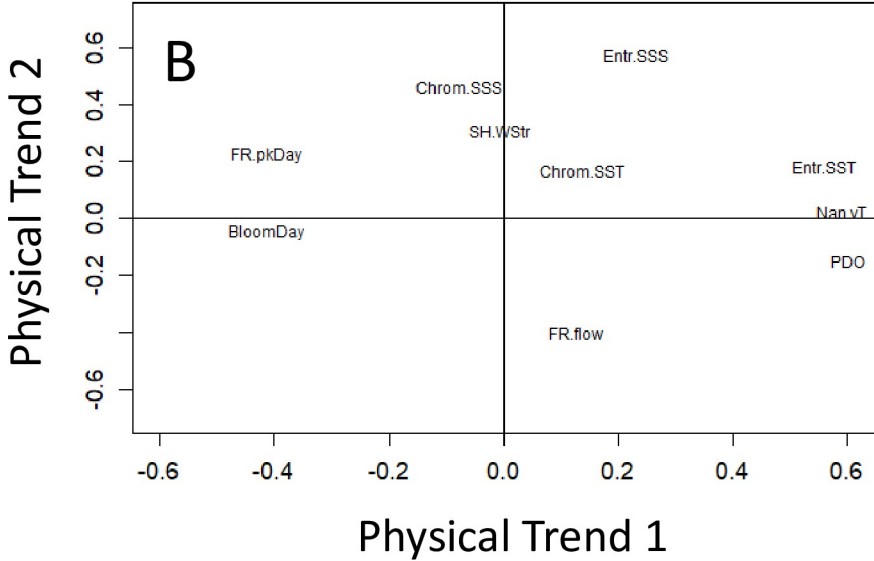

**Fig 7. Loadings of the variables on latent trends 1 and 2.** A) The 12 zooplankton taxonomic groups, B) the 10 physical variables. Abbreviated variable names are defined in Table 1.

when the complete time series was evaluated directly (1996 to 2018). However, the results from the chronological clustering (Fig 8B) indicated a change in conditions between 2006 and 2007. We therefore reran our regression models with the variables separated into two time periods: 1996 to 2006, and 2007 to 2018. This produced significant results (Table 5). The first half (the decline from 1996 to 2006) of zooplankton trend 1 was significantly related to the total annual flow from the Fraser River, the Pacific Decadal Oscillation, the sea surface salinity at Entrance Island, and the peak day of the spring phytoplankton bloom. The second half (the increase from 2007 to 2018) was also significantly related to the sea surface salinity at Entrance Island and the peak day of the spring phytoplankton bloom (Table 5). During the period from

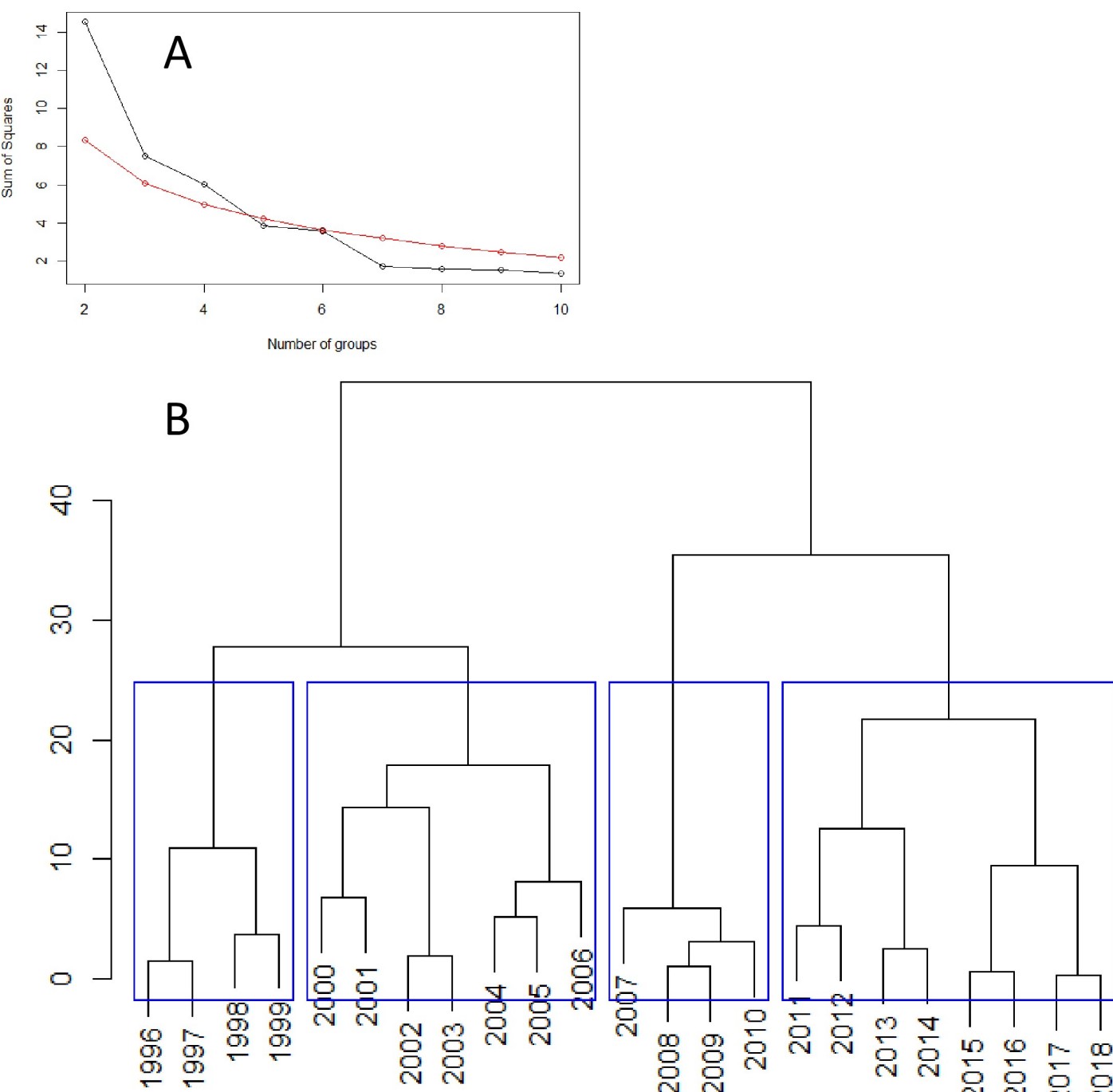

**Fig 8. Constrained (chronological) clustering based on the two zooplankton and the two physical latent trends.** A) Broken stick model to derive the number of significant clusters (black values on or above the red line are significant). B) Cluster dendrogram. Y-axis represents the Euclidean distance (magnitude of the difference) between successive comparisons of years. Blue boxes represent the four statistically-significant clusters.

1996 to 2006, zooplankton trend 2 was related to the day of the peak flow in the Fraser River and to the PDO. During the period from 2007 to 2018, zooplankton trend 2 was related to the sea surface temperature at Entrance Island and to the PDO (Table 5). The Pacific Decadal Oscillation, therefore, was significantly related to zooplankton trend 2 during both time

**Table 5. Best multiple regression models of the two zooplankton latent trends (Z1, Z2) and annual anomalies of total zooplankton biomass (TB), against the 10 physical variables.**

| Dependent variable | | Z1 (1996–2006) | Z1 (2007–2018) | Z2 (1996–2006) | Z2 (2007–2018) | TB (1996–2006) | TB (2007–2018) |
|---|---|---|---|---|---|---|---|
| **KPSS test (P-value)** | Trend | >0.1 | >0.1 | 0.056 | >0.1 | >0.1 | >0.1 |
| | Level | 0.07 | 0.051 | >0.1 | 0.053 | 0.1 | >0.1 |
| **Regression estimates for independent variables** | Nan.vT | | | | | | |
| | FR.pkDay | | | 1.05 | | 0.14 | |
| | FR.flow | -3.55 | | | | | |
| | SH.WStr | | | | | | |
| | Entr.SST | | | | 1.95 | | |
| | Entr.SSS | -5.84 | -2.62 | | | | -0.12 |
| | Chrom.SST | | | | | | |
| | Chrom.SSS | | | | | -0.29 | |
| | PDO | -0.72 | | 0.64 | -1.04 | | |
| | BloomDay | -0.54 | -2.15 | | | | |
| **ARIMA model** | Order (p,d,q) | (0,0,0) | (0,0,0) | (0,0,0) | (0,0,0) | (0,0,0) | (0,0,0) |
| | AICc | 38.15 | 35.77 | 38.26 | 22.63 | -7.13 | -20.3 |
| | Ljung-Box test for autocorrelation in residuals (P value) | 0.067 | 0.18 | 0.12 | 0.53 | 0.74 | 0.45 |
| **OLS regression** | $_{adj}R^2$ | 0.87 | 0.79 | 0.52 | 0.76 | 0.53 | 0.29 |
| | P value | 0.002 | <0.001 | 0.02 | <0.001 | 0.02 | 0.04 |

Other variable names are defined in Table 1. "KPSS test" refers to the P-value derived from the Kwiatkowski-Phillips-Schmidt-Shin test for either trend or level stationarity of the dependent time series.

periods, and with zooplankton trend 1 during the 1996–2006 time period (Table 5). The sea surface salinity at Entrance Island and the peak day for the spring phytoplankton bloom were significantly related to zooplankton trend 1 in both time periods (Table 5).

Total zooplankton biomass was also modelled as a function of the 10 physical variables. As with the two latent zooplankton trends, we separated the annual anomalies of total zooplankton biomass into the same two time periods, 1996 to 2006, and 2007 to 2018. For the period 1996 to 2006, total biomass was significantly related to the day of peak flow in the Fraser River and the sea surface salinity at Chrome Island (Table 5). For the period 2007 to 2018, total biomass was significantly related only to the sea surface salinity measured at Entrance Island (Table 5). Note that the sea surface salinity at Entrance and Chrome Islands both loaded positively on physical trend 2, indicating they had weakly similar (correlation R = 0.43, S3 Table) patterns of variability. We interpret this to mean that total zooplankton biomass was related (negatively, Table 5) to salinity in both time periods.

## Chinook and Coho salmon marine survivals

The trends in marine survival of the four salmon populations examined in our study differed over the period of ocean entry years 1996 to 2015 (S2 Fig). Two of the populations, Cowichan River Chinook and Big Qualicum Coho, had lower marine survivals during the mid 2000s. The Puntledge River Chinook population had no trend although with higher survivals at both the beginning and end of this period. Early marine survivals for Harrison River Chinook had an increasing trend (S2 Fig).

The statistical fits derived from the multiple linear regressions and ARIMA modelling of the early marine survival rates of Chinook salmon against the zooplankton and physical

**Table 6. Best statistical models describing early marine survival rates for selected stocks of Chinook and Coho salmon entering the Strait of Georgia as juveniles, against the available zooplankton and physical variables.**

| Stock | Model | KPSS test (P-value) | | Linear regression model | | | ARIMA model | | |
|---|---|---|---|---|---|---|---|---|---|
| | | Trend | Level | Model $_{adj}R^2$ | Model P-value | Cross-validation $R^2$ | Model order (p,d,q) | AICc | Ljung-Box test for autocorrelation in residuals (P value) |
| **Chinook** | | | | | | | | | |
| Cowichan River | $\alpha$ = -0.354** + -1.107*Entr.SSS*** + 1.579*Chrom.SST**+ 3.368*TotBiom*** | >0.1 | >0.1 | 0.85 | <<0.001 | 0.86 | (0,0,0) | 28.13 | 0.13 |
| | $\beta$ = ($\alpha$ * 0.53) - 4.82 | | | | | | | | |
| | S = exp($\beta$)/(exp($\beta$)+1) | | | | | | | | |
| Puntledge River | $\alpha$ = -0.013 + 1.123*Entr.SSS* + 5.505*CalCops.med** | >0.1 | >0.1 | 0.38 | 0.008 | 0.52 | (0,0,0) | 49.17 | 0.29 |
| | $\beta$ = ($\alpha$ * 0.50)– 4.97 | | | | | | | | |
| | S = exp($\beta$)/(exp($\beta$)+1) | | | | | | | | |
| Harrison River | $\alpha$ = 0.007 + 1.629*FR.flow** + 6.615*SH.WStr** + 0.962*Fish*** + 2.120*Chaetognatha* | >0.1 | 0.06 | 0.56 | 0.003 | 0.62 | (0,0,0) | 46.89 | 0.60 |
| | $\beta$ = ($\alpha$ * 0.72)– 4.02 | | | | | | | | |
| | S = exp($\beta$)/(exp($\beta$)+1) | | | | | | | | |
| **Coho** | | | | | | | | | |
| Big Qualicum River | $\alpha$ = -0.313. + 3.251*TotBiom** + 1.732*CalCops.smal* | 0.06 | >0.1 | 0.55 | 0.0002 | 0.63 | (0,0,0) | 51.96 | 0.44 |
| | $\beta$ = ($\alpha$ * 0.89)– 4.80 | | | | | | | | |
| | S = exp($\beta$)/(exp($\beta$)+1) | | | | | | | | |

'Model' refers to the regression model built with the salmon marine survival rate as the dependent variable. Superscript characters after each model parameter indicate the probability that the value is equal to 0 (*** $P<0.001$, ** $0.001 < P < 0.01$, * $0.01 < P < 0.05$, $0.05 < P < 0.1$). To back-calculate from the regression model to the original marine survival data requires reversing the scaling and the logit-transformation: these equations are provided as $\beta$ and "S" under the description of the model, in which "S" refers to the salmon marine survival rate in its original units. Variable abbreviations are defined in Table 1. "KPSS test" refers to the P-value derived from the Kwiatkowski-Phillips-Schmidt-Shin test for either trend or level stationarity of the dependent time series.

variables were good (Table 6, Fig 9). The amount of variance explained (adjusted $R^2$) ranged from 0.38 to 0.85, with P values smaller than 0.01. With a large number of potential explanatory variables (12 zooplankton and 10 physical variables) there was a risk of overfitting the data; however, all of the best models presented in Table 6 had a small number (2 to 4) of explanatory variables. In addition, all of the models for Chinook included both physical and zooplankton variables. Removal of the zooplankton variables from these models markedly reduced the statistical fits.

The model for Cowichan River Chinook (Table 6) had three variables: annual anomalies of total zooplankton biomass (positive), annual anomalies of sea surface salinity measured at Entrance Island (negative), and annual anomalies of sea surface temperature measured at Chrome Island (positive). The model for Puntledge River Chinook had two variables: annual anomalies of medium size calanoid copepods (positive) and annual anomalies of sea surface salinity measured at Entrance Island (positive) (Table 6). The model for Harrison River Chinook early marine survival had four variables (Table 6): annual anomalies of larval fish biomass (positive), annual anomalies of chaetognath biomass (positive), annual anomalies of flow volumes from the Fraser River (positive), and annual anomalies of the wind stress measured at Sand Heads at the mouth of the Fraser River (positive).

Prediction skill of these models ($R^2$, derived from 5 repeat 5-fold cross-validation) was relatively high, ranging from 0.52 to 0.86 (Table 6). The 95% confidence intervals of the predicted early marine survival rates for years with incomplete returns (ocean entry years 2016 to 2018) included the preliminary estimates for all three Chinook populations, except for Cowichan Chinook in 2018 (Fig 9).

The model for the total marine survival of the Big Qualicum Coho population (Fig 10) had similar statistics to those for the three Chinook populations and included two variables: annual biomass anomalies of total zooplankton and small calanoid copepods (Table 6). This was the only salmon model in our study which was entirely based on zooplankton variables.

## Discussion

Within the years examined in this study (1996–2018), the mid 2000s was a period of low biomass for most (70%) of the zooplankton taxonomic groups sampled in the deep areas of the Strait of Georgia. This was apparent in the patterns for total zooplankton biomass and zooplankton trend 1. After 2011, half of the zooplankton groups in this study had increased to series high biomass anomalies. This included taxa represented by zooplankton trend 1 that had recovered from their minima in the mid 2000s, and taxa represented by zooplankton trend 2 that had increasing biomass anomalies over the study period. The most significant separation in the chronological clustering, which was derived from the latent zooplankton and physical trends, occurred between 2006 and 2007. Over the period 1996 to 2018, three of the four salmon populations in this study (Cowichan Chinook, Harrison Chinook, Big Qualicum Coho) also had increasing early marine survivals after 2005. Taken together, these results suggest the plankton biomass (and therefore potentially the productivity, although this was not tested directly in this study) had increased since the late 2000s in the Strait of Georgia.

Total zooplankton biomass was dominated by a few taxa with larger individuals, such as euphausiids, large and medium sized calanoid copepods, and amphipods. All of these groups had low biomass anomalies in the mid 2000s, and most had recovered only to their climatological (1996–2010) means by 2012. For example, large calanoid copepods made up 19% of the annual total zooplankton biomass between 1996 and 2006, but only 9% of the annual total zooplankton biomass from 2007 to 2018. In contrast, slightly over half (11 out of 20) of the zooplankton groups examined in this study had their highest biomass anomalies after 2011 (e.g. hyperiid amphipods, meroplanktonic larvae of benthic animals, small calanoid copepods, ctenophores, non-calanoid copepods, Repantia (crab larvae), and siphonophores; S1 Fig). Total zooplankton biomass, therefore, was not a good indicator of biomass trends for the entire zooplankton community in the deep water areas of the Strait of Georgia.

There is a clear interpretation of the two latent physical trends. Physical trend 1 presented a sinusoidal pattern with about a ten year periodicity. This trend represented a sea temperature and event timing axis. The Pacific Decadal Oscillation loaded strongly positive on this axis and was correlated with the sea surface temperature at Entrance Island (r = 0.6, S3 Table). As its name implies, the PDO has an approximately decadal periodicity, and is considered to be the dominant year-round pattern of monthly sea surface temperature variability in the oceanic North Pacific [45]. It also influences the Strait of Georgia [46]. Newman et al. [45] concluded, however, that the PDO is not a single phenomenon but represents the interactions among a number of remote and local atmospheric and oceanic processes acting on different time scales. It therefore can serve as a proxy for, and in our data was positively correlated (r>|0.6|) with, other atmospheric indices such as those relating to El Niño–Southern Oscillation events (e.g. ONI, MEI, SOI). It was also correlated (negatively) with the North Pacific gyre circulation and regional scale upwelling, as represented by the NPGO. The peak date for the spring

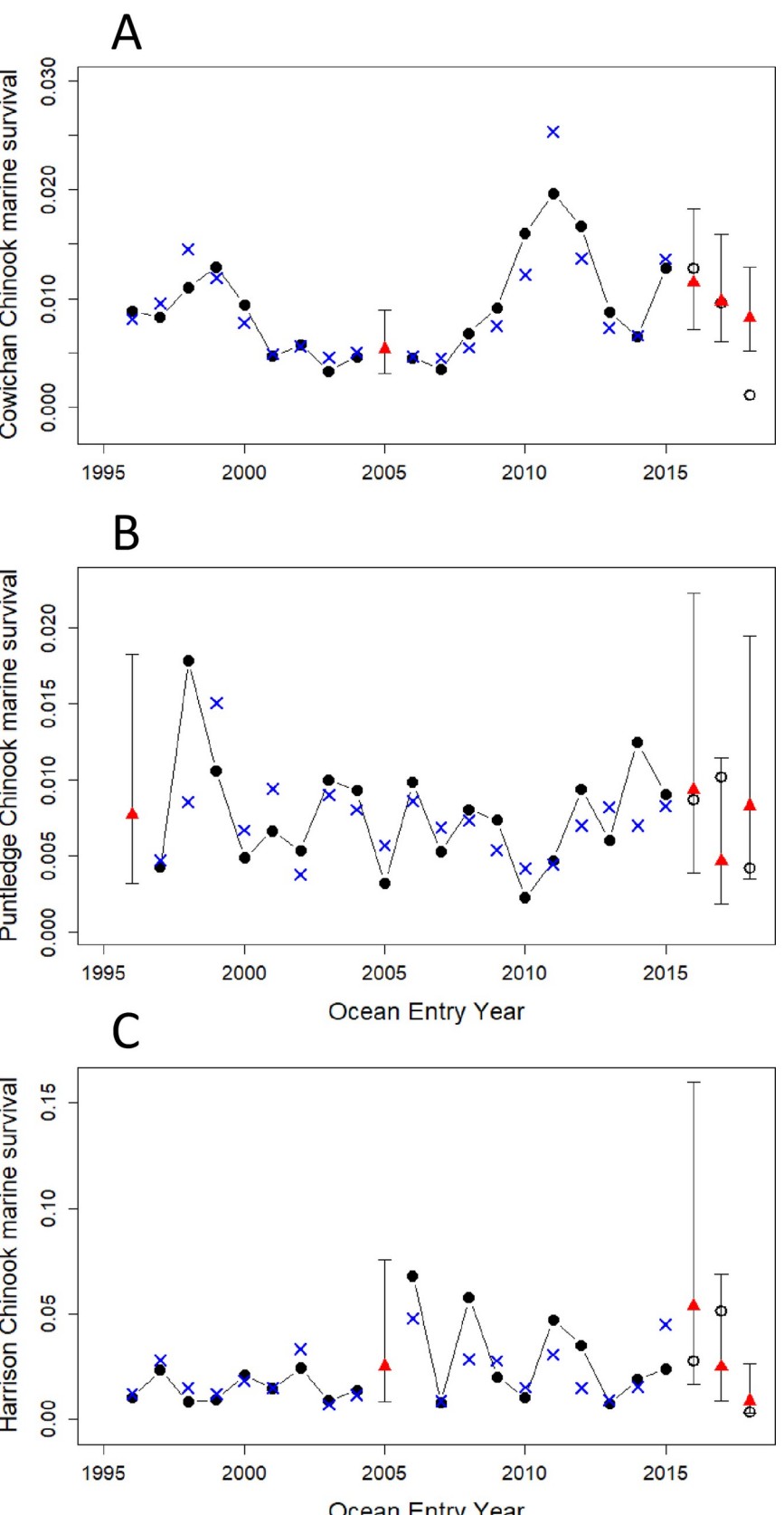

**Fig 9. Statistical model fits against original early marine survival rates for Chinook salmon stocks which enter the Strait of Georgia as smolts.** Black dots and line: original marine survival data; open black circles: estimated survivals from incomplete returns; blue crosses: model fits; red triangles and vertical lines: predicted marine survivals and their 95% confidence intervals using the model with its explanatory variables. A) Cowichan River Chinook, B) Puntledge River Chinook, C) Harrison River Chinook stocks. Note that Y-axis scales differ. Model statistics are provided in Table 6.

phytoplankton bloom and the day of peak flow in the Fraser River represented the event timing variables on physical trend 1. Physical trend 2 represented a sea surface salinity and vertical stratification axis. It represented high surface salinities in the early 2000s which declined after 2010. These two latent physical trends, therefore, present a picture of generally increasing temperatures (although with some decadal cycling) and decreasing salinity in the Strait of Georgia since the early 2000s, consistent with other analyses [16].

Six physical variables were significant in the statistical modelling of the two latent zooplankton trends when separated into 1996–2006 and 2007–2018 periods. The number of significant statistical relationships in which each variable occurred (to a maximum of 4) was: PDO (3), spring bloom date (2), sea surface salinity at Entrance Island (2), sea surface temperature at Entrance Island (1), and the volume of flow (1) and the day of peak flow (1) from the Fraser River. The U-shaped pattern of zooplankton trend 1 was significantly (and negatively) related

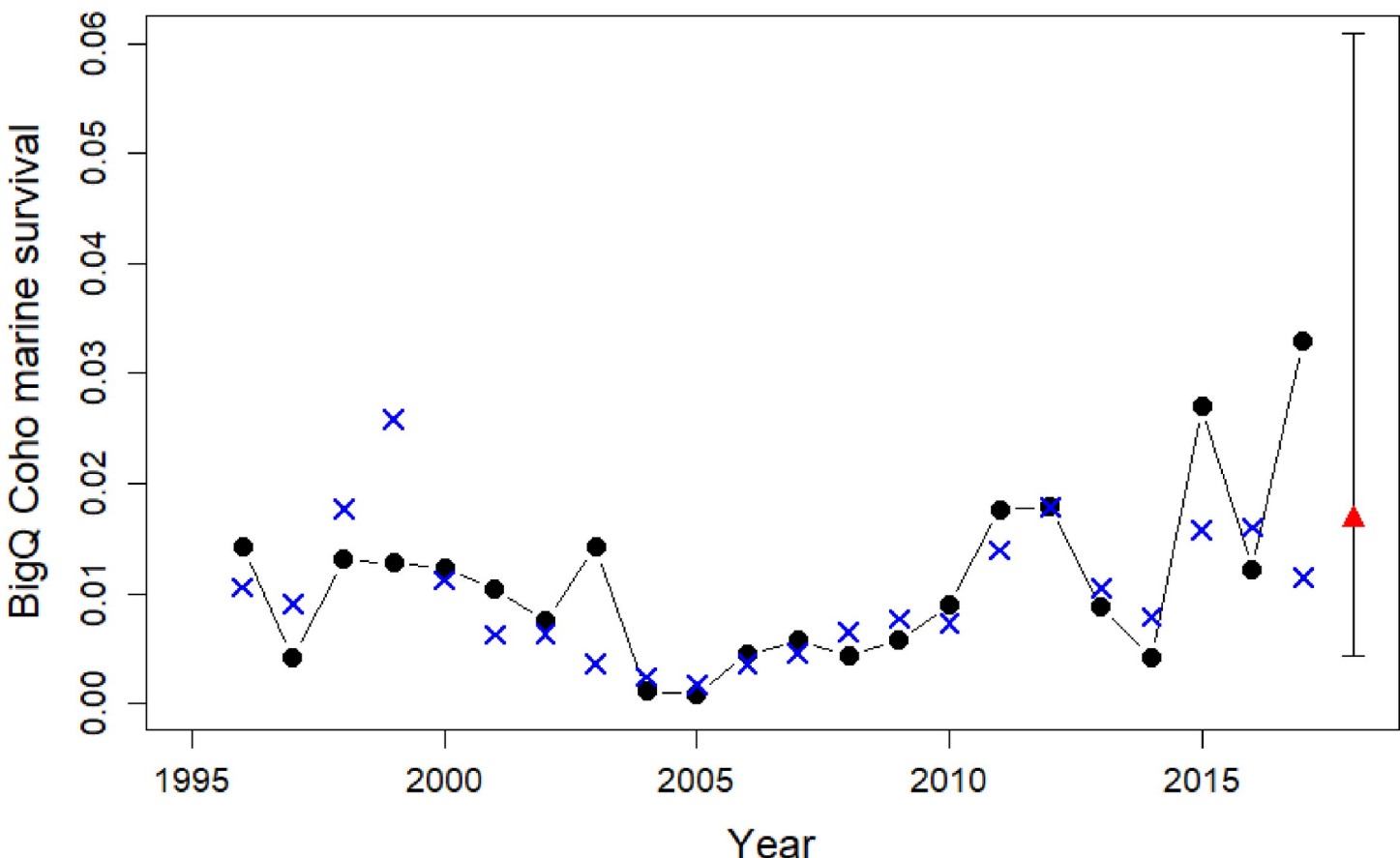

**Fig 10. Statistical model fits against original marine survival rates for the Big Qualicum Coho stock, which enter the Strait of Georgia as smolts.** Black dots and line: original marine survival data; blue crosses: model fits; red triangles and vertical lines: predicted marine survival and its 95% confidence interval using the model with its explanatory variables. Model statistics are provided in Table 6.

to the salinity at Entrance Island and the date of the spring phytoplankton bloom. Zooplankton trend 2 was significantly related to the PDO, sea surface temperature at Entrance Island, and the day of peak flow in the Fraser River. Three physical variables were significantly related to the total zooplankton biomass, representing salinity and the timing of fresh water input from the Fraser River (sea surface salinities from Entrance and Chrome Islands, and the date of peak flow in the Fraser River). Salinity (inversely) and the amount and timing of fresh water flowing into the Strait of Georgia were related to the patterns of zooplankton trends 1 and 2, and to total zooplankton biomass, in both time periods.

Early marine survivals for Chinook and total marine survivals for Coho salmon populations which enter the Strait of Georgia as juveniles have varied considerably since the 1980s. Marine survival rates were higher during the late 1980s and early 1990s than during the 2000s [9, 10]. Understanding the causes of the strong declines in Chinook and Coho marine survival in the early 1990s is one of the objectives of the Salish Sea Marine Survival Program. Our high-quality zooplankton data are available only since 1996, therefore our analyses are unable to provide direct information on this major decline. Our data can, however, provide information on the variability of Chinook and Coho salmon populations since 1996.

There were strong statistical relationships between the annual anomalies of these zooplankton and physical variables and the annual early marine survivals of several Chinook and Coho salmon populations which enter into the Strait of Georgia. Each relationship had relatively high explanatory power, and high predictive performance. These results lend support to other studies which have suggested that local ocean conditions during out-migrations to marine waters can have important effects on survival for Chinook and other salmon populations (e.g. [47–49]). All of the relationships had only a small number of variables, and included at least one zooplankton variable. Juvenile Chinook salmon in the Strait of Georgia feed upon larger zooplankton, such as euphausiids, amphipods, larval decapods, and larval and juvenile fish [50, 51]. The inclusion of total zooplankton biomass in the relationship with Cowichan River Chinook, of medium sized calanoid copepods in the relationship with Puntledge River Chinook, and of larval fish in the relationship with Harrison River Chinook, is therefore consistent with known key prey items for Chinook salmon in this region. These three Chinook populations enter the Strait of Georgia in different locations (Cowichan and Harrison enter the central Strait, whereas Puntledge enters the northern Strait of Georgia), and at different times (Harrison Chinook enter the marine environment in the summer whereas the other populations enter in late spring [52]). Harrison River Chinook also have a different marine survival pattern, and often higher marine survival rates, than the other two Chinook populations considered in this study [9].

No physical variables were significant in the minimum regression model for Big Qualicum Coho salmon. Its relationship with total zooplankton biomass (the more significant variable) may represent a direct connection with those taxa comprising the largest proportion of total biomass, such as euphausiids and amphipods (e.g. [53]), whereas the relationship with small copepods may represent an indirect effect through the food web (for example as food for amphipods [54]). A Bayesian network analysis of Coho marine survival (not separated to hatchery origin) in the Strait of Georgia over the period 1997 to 2007 also found the total zooplankton biomass anomaly and calanoid copepod biomass to be the top two predictors, followed by Pacific herring biomass and water temperature [55]. Note that since Coho normally spend 17 months at sea, part of which may include time outside the Strait of Georgia along the west coast of Vancouver Island, several other factors can influence their total marine survival (the variable used in this study). These include pinniped predation on both smolts and returning adults (and changes in pinniped behaviour), lack of estuarine food sources and habitats,

increases in predatory piscine species, and lower quality (e.g. essential fatty acids) of plankton due to higher water temperatures. These factors were not examined in this study.

On their own, large calanoid copepods were not a significant predictor of salmon early marine survival patterns in the Strait of Georgia. Among the large calanoid copepod group, *Neocalanus plumchrus* dominated the biomass in the 1990s and from 2011–2013 but were low otherwise. *Eucalanus bungii* typically had low biomass throughout the study period and was equal to or greater than *N. plumchrus* in the later 2000s and in 2017 and 2018. Typically, *N. plumchrus* occurs in the upper 100 m of the Strait of Georgia during early spring (and is deeper at other times of the year), and is replaced in this depth range by *Eucalanus* in late spring and summer [14]. *E. bungii* is more transparent than *N. plumchrus*, which may make it a more difficult prey for visual predators, such as juvenile salmonids, to capture. It also may be less nutritious than *N. plumchrus* in the Strait of Georgia, in terms of its fatty acid composition which is essential for fish growth [56]. Vertical life cycle migrations and timing, difficulty of capture, and poorer nutritional value may have combined to reduce the importance of large calanoid copepods for juvenile Chinook and Coho early marine survival in this region.

Sharma et al. [49] found that sea surface temperature was significantly related to the early marine survival of several Northeast Pacific Chinook populations; temperature also influences physiological rate processes (e.g. [57]). The presence of temperature in the regression relationship with Cowichan River Chinook salmon was therefore expected. The prominence of salinity, and other measures of fresh water flow from the Fraser River, in the statistical relationships with the latent zooplankton trends, total zooplankton biomass, and with the three Chinook salmon populations, was unexpected. In temperate estuarine systems such as the Strait of Georgia, it is salinity which controls the density field and therefore the hydrodynamics, with temperature largely a passive scalar [58]. Stronger fresh water inputs from the Fraser (and other river systems) enhance the circulation of the Strait of Georgia due to the steric effects of density, and increase the rate of flushing out (and return flow in) through the Strait of Juan de Fuca. In the context of the regression relationships which included total zooplankton biomass, therefore, lower salinities should lead to increased biomass by direct advection of zooplankton from the west coast of Vancouver Island or increased supply of nutrients. The salinity at Entrance Island was a strong contributor to the regression relationships for Cowichan and Puntledge River Chinook salmon early marine survival, but the signs of the relationships differed (negative for Cowichan, positive for Puntledge). This suggests that salinity may act differently on the processes affecting the early marine survival of Chinook salmon between the central and northern Strait of Georgia. In the Central Strait, lower salinities may positively affect the larger zooplankton and the food web for Chinook salmon by causing stronger horizontal flushing and replenishment. In the northern Strait, higher salinities may positively affect zooplankton production by providing greater vertical mixing thereby improving conditions for growth. In addition, circulation modelling studies have found the northern and southern Strait of Georgia are only weakly connected, with simulated drift particles generally remaining in their region of release [59].

Considering the short number of years and the statistical (rather than mechanistic) nature of these models, they may not be very useful for forecasting future early marine survivals (although some may be more useful than others, e.g. Cowichan River Chinook, which had reasonable 95% confidence intervals). However, they may be useful for identifying 'bottom-up' processes that influence salmon early marine survival, and therefore useful to identify years for which these processes may have had less influence on early marine survival. Large residuals between modelled and observed survivals may indicate years in which the 'usual bottom-up' mechanisms controlling early marine survival (at least as represented by these simple models) broke down, i.e. years in which other factors may have dominated marine survival. Among the

Chinook and Coho populations examined in this study, years which appeared to have larger residuals included 1996, 1997, 2002, 2003, and 2017 (Figs 9 and 10; specific stocks have additional years with large residuals). Such other factors may include physical processes not in these models, and predation, for example by pinnipeds (e.g. [12]).

Our results are consistent with those of Mackas et al. [14], who did a similar analysis using data from 1990 to 2010 (although they did not consider relationships with Chinook salmon). They found three principle components described the trends in their selection of zooplankton data. Their trend 1 had a similar pattern to our trend 1. In our analysis, trend 1 continued to increase from 2010 to 2018. Trend 2 from Mackas et al. [14] had a generally similar, although inverse, pattern to our zooplankton trend 2. Mackas et al. [14] found the North Pacific Gyre Oscillation (NPGO) Index was the strongest environmental predictor of zooplankton biomass interannual variability, followed by winter winds (affecting the timing of the spring phytoplankton bloom) and the Southern Oscillation Index (SOI). Li et al. [15] also found the SOI to be the best predictor of zooplankton composition change for a subset of these data. In our results, the PDO was significant in three of the four regressions of the latent trends against physical variables; both NPGO and SOI were removed from our statistical modelling because they were highly correlated (r>|0.60|) with the PDO. In addition, the date for the spring phytoplankton bloom was significant in two of the four regression models of the latent zooplankton trends against the physical variables. In contrast to Mackas et al. [14], who used data from 1990 to 1996 despite their poorer taxonomic resolution, we chose to exclude these data. Our study also included samples from 2011 to 2018, which were not available to Mackas et al. [14]. These years clustered separately from adjacent years suggesting that recent conditions had changed from the range of years studied by Mackas et al. [14].

In this study, we did not examine the within-year seasonal or monthly changes that occurred among the zooplankton and the physical variables, which can be considerable (this will be the subject of a further study). In addition, it is worth noting that salmon generally are surface-oriented, with juveniles in coastal waters usually occurring between the surface and 15 m [60], and that they feed mostly at dawn or dusk [61]. In contrast, our zooplankton tows covered the full water column during both daylight and night. Such a tow pattern was necessary to produce a consistent series of zooplankton data over time, unaffected by variations in the depth range of the tow. As a consequence, however, not all the zooplankton that we sampled were necessarily available to the juvenile salmon, although with vertical migration of the zooplankton most will be in the upper water layers at dawn and dusk when juvenile salmon are actively feeding. We are unable, therefore, to definitively conclude whether the relationships identified in this study are a result of direct effects of zooplankton on salmon, or indirect effects due to correlations of both zooplankton and salmon with other variables. Further, the methods we used to identify the relationships were all statistical rather than mechanistic. In particular, we did not consider potential non-linear effects. Therefore, if conditions change, the relationships derived from this study may lose their predictive skill (e.g. [62, 63]). We also did not consider time lags among the relationships between physical and zooplankton variables, as our dependent data (annual zooplankton biomasses) were already limited (23 years). However, the generally strong statistical relationships are useful as syntheses of the patterns among the physical variables, zooplankton biomass, and early marine survivals of several Chinook and Coho salmon populations in the Strait of Georgia.

One goal of community ecology is to identify the processes controlling the dynamics underlying the observed patterns of abundance and distributions of organisms. Bakun [64, 65] proposed three key processes governing the productivity of marine populations: enrichment, concentration, and retention (collectively referred to as "Bakun's Triad"). Masson and Perry [66] proposed that these processes are central to the dynamics of the Strait of Georgia marine

ecosystem, and added three biologically-related processes: initiation of plankton blooms, trophic dynamics, and nearshore/benthic dynamics. Modelling is the definitive tool to assess how these processes interact with the responses of individual, and groups of, species to their biological and physical environments. We have begun working with researchers at the University of British Columbia to understand the processes driving zooplankton populations in the Strait of Georgia (e.g. [67]). The correlations of the latent zooplankton trends with sea surface salinity and flow from the Fraser River suggest the retention process of Bakun's Triad would be a good initial focus for these modelling studies. Other studies (e.g. [68]) have also suggested that zooplankton community composition in this region is determined by currents interacting with zooplankton vertical migration behaviours, among other processes.

## Conclusions

The availability of long time series of zooplankton data from the Strait of Georgia provides a means to understand the natural variability of this marine ecosystem. Our results indicate that, at least within the time period of this study, interannual variability in marine survivals of three Chinook and one Coho salmon population in the Strait of Georgia were controlled by biophysical, rather than human, processes. Annual total zooplankton biomass anomalies were highest in the late 1990s, lowest in the mid 2000s, and have been generally above their climatological (1996–2010) average since 2011. Four zooplankton groups comprised 76% of the total zooplankton biomass, on average, each year: euphausiids, large and medium sized calanoid copepods, and amphipods. However, half of the zooplankton taxonomic groups examined had their highest biomass anomalies after 2011, including hyperiid amphipods, meroplanktonic larvae of benthic animals, small calanoid copepods, ctenophores and siphonophores. Most of these groups consisted of small-sized individuals or were not very abundant, and so were not major contributors to the total biomass. Total zooplankton biomass does not appear to be a good indicator of biomass trends for all zooplankton taxonomic groups in the Strait of Georgia.

Two latent (underlying) trends in the zooplankton data were identified: taxa with a U-shaped pattern with a minimum in 2009, representing seven out of 12 groups, and a decline from 2003 to 2011, characterising the patterns of variability of chaetognaths (positive) and fish larvae, medusae, and ctenophores (negative, meaning these groups increased over these years without a minimum in the mid-2000s). Two latent trends were also identified from the 10 physical variables retained in the analysis. Physical trend 1 represented a sea temperature and event timing (spring bloom peak, day of peak flow in the Fraser River) axis, and trend 2 represented a sea surface salinity and vertical stratification axis. The physical variables significantly related to the two latent zooplankton trends were the Pacific Decadal Oscillation, sea surface salinity measured at Entrance Island in the central Strait of Georgia, and the (modelled) peak date of the spring phytoplankton bloom in the central Strait of Georgia. The variability of these zooplankton and physical variables clustered into two distinct periods: 1996–2006, and 2007–2018, representing declining zooplankton biomass anomalies followed by increasing zooplankton biomass anomalies.

Two of the four salmon populations also had minima in their marine survival rates in the mid 2000s: Cowichan River Chinook and Big Qualicum Coho. A small number (2 to 4) of explanatory zooplankton and physical variables were significant in models of the early marine survival rates for the three Chinook salmon populations (Cowichan River, Puntledge River, and Harrison River), and the total marine survival of the Big Qualicum Coho population. All relationships included zooplankton variables; prominent among the physical variables were sea surface salinity and other physical variables representing the flow from the Fraser River.

These regression relationships explained ($_{adj}R^2$) from 38 to 85% of the annual variability in marine survival rates of these four salmon populations. A consistent zooplankton monitoring program in the Strait of Georgia and the broader Salish Sea would help with rapid identification of important changes in this system and could assist with projections of future salmon marine survivals.

## Supporting information

**S1 Table. Definitions of which zooplankton taxa were included in each taxonomic group classification.** Abbreviated group names are defined in Table 1. F = Female; M = Male; s1 = size class 1 (<2 mm); s2 = size class 2 (2–5 mm); s3 = size class 3 (>5 mm). For copepods, numbers refer to life history stage (with stage 6 being the reproducing adult). '* sp.' means not identified to species.
(XLSX)

**S2 Table. Excel file with annual anomalies of the 20 zooplankton groups and the 16 physical variables initially considered in this study, and the salmon marine survival rates (year is ocean entry year).** Anomalies have been generated using the methods described in the text. Abbreviations are defined in Table 1 in the text.
(XLSX)

**S3 Table. Excel file with correlations among the annual anomalies of the 20 zooplankton taxonomic groups and the 16 physical variables, initially considered in this study.** Abbreviations are defined in Table 1; peach coloured cells are those with correlations greater than 0.60, blue coloured cells have negative correlations greater than -0.60.
(XLSX)

**S1 Fig. Annual anomalies of $\log_{10}$ biomass (g m$^{-2}$) of the 20 zooplankton taxonomic groups, from 1996 to 2018.** Baseline (climatology) period was 1996–2010. Y-axes are the annual anomalies; note the scales differ. Y-axes are labelled with the abbreviated names of each zooplankton group (abbreviated names are defined in Table 1).
(PDF)

**S2 Fig. Transformed and scaled salmon early marine survival patterns.** (A) Cowichan River Chinook, (B) Harrison River Chinook, (C) Puntledge River Chinook, and (D) Big Qualicum Coho salmon populations for ocean entry years 1996 to 2015 for Chinook, and to 2017 for Coho. Black dots and lines represent the observed early (for Chinook) or total (for Coho) marine survivals. Blue line and shading represent 95% confidence bands about a loess smoother applied to these annual values (derived from a general additive model with year as the independent variable) to better show the general patterns.
(PDF)

**S1 File. Numerical simulation to examine the effect of uneven sampling among years.**
(PDF)

## Acknowledgments

We thank the Pacific Salmon Foundation, and Dr. Brain Riddell and Dr. Isobel Pearsall, for their interest in this work. We are especially grateful to the crews of the MV *Salacia*, CCGS *John P. Tully*, CCGS *Vector*, and CCGS *Neocaligus* for their help at sea, as well as the support of many DFO technical and administrative leadership and staff, in particular Dr. David Mackas who started the program, Mark Belton, Glenn Cooper, Marie Robert, and Doug

Yelland. We also thank Dr. Susan Allen of the University of British Columbia for discussions of the physical dynamics of the Salish Sea. The comments from two anonymous reviewers helped substantially to improve this manuscript. This is contribution number 51 to the Pacific Salmon Foundation / Long Live the Kings Salish Sea Marine Survival Project.

## Author Contributions

**Conceptualization:** R. Ian Perry, Kelly Young, Moira Galbraith.

**Data curation:** R. Ian Perry, Kelly Young, Moira Galbraith, Peter Chandler, Antonio Velez-Espino, Steve Baillie.

**Formal analysis:** R. Ian Perry, Kelly Young, Moira Galbraith.

**Funding acquisition:** R. Ian Perry.

**Investigation:** Kelly Young, Moira Galbraith.

**Methodology:** R. Ian Perry, Kelly Young, Moira Galbraith, Peter Chandler, Antonio Velez-Espino, Steve Baillie.

**Project administration:** R. Ian Perry.

**Software:** R. Ian Perry, Kelly Young, Moira Galbraith, Antonio Velez-Espino, Steve Baillie.

**Supervision:** R. Ian Perry.

**Validation:** R. Ian Perry, Kelly Young, Moira Galbraith, Peter Chandler, Antonio Velez-Espino, Steve Baillie.

**Visualization:** Kelly Young.

**Writing – original draft:** R. Ian Perry, Kelly Young, Moira Galbraith.

**Writing – review & editing:** R. Ian Perry, Kelly Young, Moira Galbraith, Peter Chandler, Antonio Velez-Espino, Steve Baillie.

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
