## [Decision Letter · Decision Letter 0]

10 Nov 2020

PONE-D-20-31572

Zooplankton variability in the Strait of Georgia, Canada, and relationships with the marine survivals of Chinook and Coho salmon

PLOS ONE

Dear Dr. Perry,

Thank you for submitting your manuscript to PLOS ONE. After careful consideration, we feel that it has merit but does not fully meet PLOS ONE’s publication criteria as it currently stands. Therefore, we invite you to submit a revised version of the manuscript that addresses the points raised during the review process.

Both two reviewers recognized the value of this study in a field of biological oceanography, and provided a lot of suggestions and comments for improving the ms. Please read them carefully, and rewrite it honestly.

We look forward to receiving your revised manuscript.

Kind regards,

Syuhei Ban

Academic Editor

PLOS ONE

Journal Requirements:

2. We note that Figure 1 in your submission contains map images which may be copyrighted.

We require you to either (a) present written permission from the copyright holder to publish this figure specifically under the CC BY 4.0 license, or (b) remove the figure from your submission:

b. If you are unable to obtain permission from the original copyright holder to publish this figure under the CC BY 4.0 license or if the copyright holder’s requirements are incompatible with the CC BY 4.0 license, please either i) remove the figure or ii) supply a replacement figure that complies with the CC BY 4.0 license. Please check copyright information on all replacement figures and update the figure caption with source information. If applicable, please specify in the figure caption text when a figure is similar but not identical to the original image and is therefore for illustrative purposes only.

Additional Editor Comments:

Both two reviewers recognized the value of this study in a field of biological oceanography, and provided a lot of suggestions and comments for improving the ms. Please read them carefully, and rewrite it honestly.

Reviewers' comments:

Reviewer's Responses to Questions

**Comments to the Author**

1. Is the manuscript technically sound, and do the data support the conclusions?

Reviewer #1: Yes

Reviewer #2: Yes

2. Has the statistical analysis been performed appropriately and rigorously? 

Reviewer #1: Yes

Reviewer #2: Yes

3. Have the authors made all data underlying the findings in their manuscript fully available?

Reviewer #1: No

Reviewer #2: Yes

4. Is the manuscript presented in an intelligible fashion and written in standard English?

Reviewer #1: Yes

Reviewer #2: Yes

5. Review Comments to the Author

Reviewer #1: This manuscript examines a time series of zooplankton and environmental data for the Strait of Georgia, British Columbia (Canada), using regression analysis and Dynamic Factor Analysis to identify trends and relationships between parameters. Further, the study examines the relationship of physical and biological parameters with marine survival of Coho and Chinook salmon and are able to explain a substantial amount of the variability (38-85%) in marine survival rates of select populations.

This is an exceptionally well written paper that was a pleasure to read. It will make a valuable contribution to the literature, that will be broadly relevant to the salmon community, and specifically relevant to researchers in the north-eastern Pacific. My only complaint is that the paper is somewhat descriptive, particularly with respect to the drivers if zooplankton dynamics. Although there is value in extending the existing zooplankton time series for the region, the discussion would benefit greatly from a deeper interpretation of the mechanisms controlling zooplankton dynamics.

I recommend the manuscript for publication after dealing with the above comment and the additional comments below.

Marine survivals for Coho are total marine survivals. Given that Coho may spend one to three years at sea, a number of factors other than early marine survival could be at play. Please comment on this in the discussion and consider other potential mortality agents.

Fig. 1. Please use different symbols for salmon and CTD data.

Lines 144-146: “For each taxon and stage, biomass was calculated from abundance using our database of direct measurements of local specimens or extrapolation from measured and literature values.”

As it stands, there is no way to assess the validity and quality of these biomass estimates. Recommend to include length-weight relationships in appendix / supplementary material, or a reference to where these can be validated.

Lines 749-751: “This stock originates in the Fraser River system and therefore its early marine survival can be expected to be related to conditions in the Fraser River”

I understand that marine survival for the Harrison River sock takes into account the first year at sea, so it does not necessarily follow that their marine survival is related to the Fraser River.

Reviewer #2: PONE-D-20-31572

Review

The paper Zooplankton variability in the Strait of Georgia, Canada, and relationships with the marine survivals of Chinook and Coho salmon reports on an investigation of the environmental factors driving the zooplankton community in the Strait of Georgia and relates underlying physical drivers and zooplankton abundances with Chinook and coho salmon marine survival. This paper represents a large undertaking—from collection to taxonomy to modeling and integrated description—and provides a needed rigorous assessment of the lower trophic levels. As the authors note, their work is statistically based, not mechanistic, but the statistical treatment is robust and provides an indication of both the changes in zooplankton community over time and the physical variables of importance. The paper has the added plus of drawing the connection between the plankton community and salmon species of concern in the Salish Sea.

Overall, the paper is well-written and synthesizes a large amount of information clearly. I note only a few concerns. The description of the statistical methods was generally easy to follow, but some additional detail could be provided for those less familiar with the numerous methods (and I will acknowledge familiarity and understanding of all the methods used, but have only employed a subset of them on my own to the point of being intimately familiar with the details). I found the description of the oceanography of the Strait of Georgia and the implications for the zooplankton community in the discussion to be particularly compelling—this merging of the disciplines (physical oceanography and biology) is a nice contribution.

Abstract

Lines 30-31: This could be rephrased for more clarity. What are the 2 physical factors? Or are these latent trends? (After reading the paper I understand, but many readers will only read the abstract—perhaps add a line about where these trends came from—DFA—and it will be clear)

Lines 28-32: Maybe the term factor is giving me problems here—I think of these DFA outputs as underlying trends, not factors…but this could just be personal preference

Intro

Line 22: Only 12 groups were used in the full analysis after removal of the colinear and sparse taxa, correct? That would seem a better number to report here.

Line 51: Is or as a recipient? Sewage outflows or just general stormwater runoff Both would apply, but the meaning is unclear.

Line 91: Variability in what within the system?

Line 100: “communities” or “patterns” might be better that “properties” to tie into obj. 1?

Methods

Line 114: A short description of these nets would be helpful to understand how they differ

Line 118: Was it rare to estimate flow with this calculation? Provide proportion of measure versus estimated samples.

Line 146: Were the literature values from local spp too? Or more general?

Line 219: How did you decide which of the correlated pairs to remove? While a different analysis, it would be interesting to know which of these taxa co-occur given certain conditions and which are responding to environmental variability differently. I suppose this can be inferred from the analysis herein, since the highly correlated taxa did in fact occur under the same conditions, but I guess I was thinking of a community analysis type approach.

Line 223: patterns

Line 227: 1-9 combinations of factors? This makes it sound as if covariates were added here, but I think you mean fitting the model with 1-9 underlying trends? As in the abstract the term “factor” seems unusual here—the MARSS manual refers to “hidden trends” or “latent trends” with “factor loadings” estimated for the individual drivers of the trends (which observed TS is +/- for the latent trends).

Line 251: N is Newton? Maybe I should know this, but don’t deal with wind much—maybe other more biologically-oriented people don’t either.

Line 305: Perhaps add “given their declines and failure to rebound” or something similar.

Lines 352-388: This is a really nice description of the methods employed. A few points needing clarification: line 354, describe the KPSS test a bit; line 369, define p,d,q; and last line, is the more complex model the one with more covariates? Or something else?

Results:

Line 438-442: This seems more appropriate for the discussion.

Line 448-449: Is there no evidence of daytime net avoidance for other species? I realize the Euphs are more susceptible because of their size and diel migration, but is this just a case of the studies not having been done for other species or do they really not exhibit it?

Line 510-ish: It is interesting to me that euphausiids did not load on either of the prevalent trends despite their dominance in the samples. This is perhaps worth noting in the text. It’s on the Fig. 7, but not explicitly mentioned.

Line 551: This is an interesting finding!

Line 554: Please explain how figure 8A demonstrates 4 groups. I would think this figure demonstrates a preference for more groups. (Ah, I see it is in the caption—maybe move to text?)

Discussion:

Line 705-706: Consider referencing recent work by Litzow et al. discussing non-stationarity and PDO in the Gulf of Alaska and how local observations are related and tying your observations to what's happening there. (https://royalsocietypublishing.org/doi/full/10.1098/rspb.2018.1855, https://esajournals.onlinelibrary.wiley.com/doi/full/10.1002/ecy.2760#support-information-section)

Line 737: We typically think of salmon as rather surface oriented, yet the zooplankton data are all collected from deeper stations. Are the zooplankton in and of themselves driving the salmon survival, or are they another indicator of some other process? Have the salmon surveys found outmigrants at the depths of the zooplankton you sampled in the Strait of Georgia? It seems that a comment on the depth relationship would help tie this together a bit better.

Line 790: I like this discussion of oceanography and production. Is there any evidence that the densities of zooplankton observed (especially the large guys) would be limiting for salmon?

Line 832: There is rather extreme uneven sampling effort in this time series and while I think good effort has been made to neutralize differences, I’m still wondering if this has bearing on the outcomes of the analysis. It would be interesting to look at the most recent years where sampling has been most robust and address variation within the samples. Surely additional samples increase the variation as the authors point out, but also they allow inference in different ways and at different scales. Perhaps other work is being done on these samples (the “further study”) that would allow for understanding community change over short time scales—this may in turn yield insight on a longer time scale. It would be nice if the authors could comment on this and whether the small sample sizes early on are “good enough” for drawing conclusions, or whether the high-intensity sampling has/is expected to contributed additional conclusions.

Line 841: This is a thorough summary but also an extremely long paragraph and could be broken up.

Line 871: This suggestion brings me back to the varying levels of effort and if there is any way to leverage that variation in determining what such a monitoring program would look like…just food for thought (if the team needs something to do).

Figs.

In figure 3, panels E and F should be pulled out and put in their own 2-panel plot so the reader has some hope of actually seeing the composition.

6. PLOS authors have the option to publish the peer review history of their article (what does this mean?). If published, this will include your full peer review and any attached files.

Reviewer #1: No

Reviewer #2: No

---

## [Author Response · Author response to Decision Letter 0]

23 Dec 2020

Dear Dr. Ban

Subject: Revised version of manuscript PONE-D-20-31572, Perry et al.

My co-authors and I wish to thank you, and the two anonymous reviewers, for your comments on our original version of our manuscript. We believe that these comments have helped to substantially improve the present version. The text below in blue represents our responses to the reviewers’ comments. 

Comments from the Editor:

We have followed the PLoS ONE file naming conventions carefully.

 2. We note that Figure 1 in your submission contains map images which may be copyrighted.

Thank you for your attention to our map (Figure 1). This figure was created by us using available computer code and bathymetry data that are freely available in the R package PBSMapping. We have added the following sentence to the caption for Figure 1, to make this clear: “Map was created using code and bathymetric data in the R package PBSmapping [18].).”

Both two reviewers recognized the value of this study in a field of biological oceanography, and provided a lot of suggestions and comments for improving the ms. Please read them carefully, and rewrite it honestly.

We appreciate the time and effort the reviewers have spent on our manuscript. We have tried to respond fully to their comments. 

Reviewers' comments:

Reviewer's Responses to Questions

Comments to the Author

1. Is the manuscript technically sound, and do the data support the conclusions?

Reviewer #1: Yes

Reviewer #2: Yes

2. Has the statistical analysis been performed appropriately and rigorously? 

Reviewer #1: Yes

Reviewer #2: Yes

3. Have the authors made all data underlying the findings in their manuscript fully available?

Reviewer #1: No Please see our response below

Reviewer #2: Yes

4. Is the manuscript presented in an intelligible fashion and written in standard English?

Reviewer #1: Yes

Reviewer #2: Yes

5. Review Comments to the Author

Reviewer #1: This manuscript examines a time series of zooplankton and environmental data for the Strait of Georgia, British Columbia (Canada), using regression analysis and Dynamic Factor Analysis to identify trends and relationships between parameters. Further, the study examines the relationship of physical and biological parameters with marine survival of Coho and Chinook salmon and are able to explain a substantial amount of the variability (38-85%) in marine survival rates of select populations.

This is an exceptionally well written paper that was a pleasure to read. It will make a valuable contribution to the literature, that will be broadly relevant to the salmon community, and specifically relevant to researchers in the north-eastern Pacific. My only complaint is that the paper is somewhat descriptive, particularly with respect to the drivers of zooplankton dynamics. Although there is value in extending the existing zooplankton time series for the region, the discussion would benefit greatly from a deeper interpretation of the mechanisms controlling zooplankton dynamics.

We thank the reviewer for her/his comments. The originally submitted version does include a discussion of the potential importance of advection and water exchange into and out of the Strait of Georgia as a processes influencing the observed zooplankton (and salmon) trends. We have provided the following paragraph just before the Conclusions section to enhance this point:

“One goal of community ecology is to identify the processes controlling the dynamics underlying the observed patterns of abundance and distributions of organisms. Bakun [64,65] proposed three key processes governing the productivity of marine populations: enrichment, concentration, and retention (collectively referred to as “Bakun’s Triad”). Masson and Perry [66] proposed that these processes are central to the dynamics of the Strait of Georgia marine ecosystem, and added three biologically-related processes: initiation of plankton blooms, trophic dynamics, and nearshore/benthic dynamics. Modelling is the definitive tool to assess how these processes interact with the responses of individual, and groups of, species to their biological and physical environments. We have begun working with researchers at the University of British Columbia to understand the processes driving zooplankton populations in the Strait of Georgia (e.g. [67]). The correlations of the latent zooplankton trends with sea surface salinity and flow from the Fraser River suggest the retention process of Bakun’s Triad would be a good initial focus for these modelling studies. Other studies (e.g. [68]) have also suggested that zooplankton community composition in this region is determined by currents interacting with zooplankton vertical migration behaviours, among other processes.”

I recommend the manuscript for publication after dealing with the above comment and the additional comments below.

Marine survivals for Coho are total marine survivals. Given that Coho may spend one to three years at sea, a number of factors other than early marine survival could be at play. Please comment on this in the discussion and consider other potential mortality agents.

A normal adult Coho spends about 17 months at sea. It is this group that is used to estimate the marine survivals. Coho that are at sea for an additional year are very rare. These Coho would be immediately identified when the coded wire tag is read and therefore are not be included in the marine survival calculation. There are innumerable sources of mortality that will affect the marine survival estimate. These include and are not limited to: pinniped predation on both smolts and returning adults, lack of estuarine food sources and habitat, loss of marine riparian zones (where wild Coho rear and transit), hazardous algal blooms, higher levels of pollutants, increased levels of disease, predatory piscine species due to higher temperature waters, and lower quality plankton due to higher temperature waters.

We have included the following sentences in the Discussion paragraph addressing Coho marine survivals:

“A Bayesian network analysis of Coho marine survival (not separated to hatchery origin) in the Strait of Georgia over the period 1997 to 2007 also found the total zooplankton biomass anomaly and calanoid copepod biomass to be the top two predictors, followed by Pacific herring biomass and water temperature [55]. Note that since Coho normally spend 17 months at sea, part of which may include time outside the Strait of Georgia along the west coast of Vancouver Island, several other factors can influence their total marine survival (the variable used in this study). These include pinniped predation on both smolts and returning adults (and changes in pinniped behaviour), lack of estuarine food sources and habitats, increases in predatory piscine species, and lower quality (e.g. essential fatty acids) of plankton due to higher water temperatures. These factors were not examined in this study.”

Fig. 1. Please use different symbols for salmon and CTD data. 

Done.

Lines 144-146: “For each taxon and stage, biomass was calculated from abundance using our database of direct measurements of local specimens or extrapolation from measured and literature values.”

As it stands, there is no way to assess the validity and quality of these biomass estimates. Recommend to include length-weight relationships in appendix / supplementary material, or a reference to where these can be validated.

In our original submission we provided (in S2 Table) the anomalies of the taxonomic groups included in this study. In that submission, we did not include data on the individual taxa and size or life history stages which were aggregated into these taxonomic groups. We apologise for this omission. In this revised version we provide directions and a link to the freely available data upon which these groups were developed. We have made the following changes to the section titled “Taxonomic analyses” in Methods:

“… The remaining subsample (containing organisms <5 mm) was split to approximately 400 individuals which were identified to the lowest possible taxonomic classification and life history stage or size class. Data were obtained from the Fisheries and Oceans Canada Institute of Ocean Sciences Zooplankton Database (version 9, data downloaded on 21 May 2020). For each taxon - stage or taxon - size class combination in each sample, biomass was calculated from the abundance and the average biomass of individuals in that sample, derived from measured or literature values (these latter may have included the same or similar species from elsewhere). These data are available on the Government of Canada open data portal under Strait of Georgia zooplankton. We used both abundance and biomass of each taxon and stage or size, calculated as numbers or weight (dry mass in grams) per unit surface area (i.e. per m2).”

We also include the direct link to this we page: “Direct link: https://open.canada.ca/data/en/dataset/2822c11d-6b6c-437e-ad65-85f584522adc”

Lines 749-751: “This stock originates in the Fraser River system and therefore its early marine survival can be expected to be related to conditions in the Fraser River”

I understand that marine survival for the Harrison River stock takes into account the first year at sea, so it does not necessarily follow that their marine survival is related to the Fraser River.

The reviewer is correct, and we have removed this sentence from the Discussion.

Reviewer #2: PONE-D-20-31572

Review

The paper Zooplankton variability in the Strait of Georgia, Canada, and relationships with the marine survivals of Chinook and Coho salmon reports on an investigation of the environmental factors driving the zooplankton community in the Strait of Georgia and relates underlying physical drivers and zooplankton abundances with Chinook and Coho salmon marine survival. This paper represents a large undertaking—from collection to taxonomy to modeling and integrated description—and provides a needed rigorous assessment of the lower trophic levels. As the authors note, their work is statistically based, not mechanistic, but the statistical treatment is robust and provides an indication of both the changes in zooplankton community over time and the physical variables of importance. The paper has the added plus of drawing the connection between the plankton community and salmon species of concern in the Salish Sea.

Overall, the paper is well-written and synthesizes a large amount of information clearly. I note only a few concerns. The description of the statistical methods was generally easy to follow, but some additional detail could be provided for those less familiar with the numerous methods (and I will acknowledge familiarity and understanding of all the methods used, but have only employed a subset of them on my own to the point of being intimately familiar with the details). I found the description of the oceanography of the Strait of Georgia and the implications for the zooplankton community in the discussion to be particularly compelling—this merging of the disciplines (physical oceanography and biology) is a nice contribution.

We thank the reviewer for her/his comments. In particular, the comment about the uneven nature of the sampling effort encouraged us to do a numerical simulation to examine this question (the results of which are discussed below). In regards to the comment on the methods, we have added additional descriptions of the statistical methods in response to the specific review comments, below.

Abstract

Lines 30-31: This could be rephrased for more clarity. What are the 2 physical factors? Or are these latent trends? (After reading the paper I understand, but many readers will only read the abstract—perhaps add a line about where these trends came from—DFA—and it will be clear)

Sentence changed to “Two latent trends (derived from dynamic factor analyses) described the variability of annual biomass anomalies underlying all zooplankton groups: a U-shaped trend with its minimum in the mid-2000s, and a declining trend from 2001 to 2011. “

 Lines 28-32: Maybe the term factor is giving me problems here—I think of these DFA outputs as underlying trends, not factors…but this could just be personal preference.

This is a great point: we have removed ‘underlying factors’ and replaced with ‘latent trends’ throughout the manuscript.

Intro

Line 22: Only 12 groups were used in the full analysis after removal of the colinear and sparse taxa, correct? That would seem a better number to report here.

Agreed. Changed accordingly.

Line 51: Is or as a recipient? Sewage outflows or just general stormwater runoff Both would apply, but the meaning is unclear.

Agreed. Changed “…is a recipient of the effluents from large and small coastal communities…”

Line 91: Variability in what within the system?

Agreed. We have changed to “… that accounted for a significant proportion of the annual patterns of variability of 37 biophysical and human data series from this ecosystem over the period 1970 to 2010…”

Line 100: “communities” or “patterns” might be better that “properties” to tie into obj. 1?

Agreed. We have replaced “properties” with “patterns”

Methods

Line 114: A short description of these nets would be helpful to understand how they differ

We have added the following to that section: “…..A ring net has a cone shape, whereas SCOR and Bongo nets have a cylindrical section at the top before tapering to the cod end. All nets are expected to have similar capture efficiencies [19]. ”

Line 118: Was it rare to estimate flow with this calculation? Provide proportion of measure versus estimated samples.

We have added the following sentence to this section:

“This occurred in less than 13% of the tows used for this analysis.”

Line 146: Were the literature values from local spp too? Or more general?

The literature values sometimes included similar species from other regions. We have added the following to that section: “…measured or literature values (these latter may have included the same or similar species from elsewhere).”

Line 219: How did you decide which of the correlated pairs to remove? 

We removed variables which were highly correlated (r>|0.60|) with several other groups, and retained variables which were highly correlated with only a few other groups. The rationale for this was that the patterns of variability represented by variables that were highly correlated with many other groups was included in those other groups, whereas the patterns of variability were (somewhat) unique in those variables which were highly correlated with only a few other groups. We have added the following to this section: “To reduce potential problems due to multi-collinearity of variables, we did a correlation analysis among all pairs of remaining zooplankton groups and removed those which were highly correlated (r >|0.60|) with several other groups (e.g. [21]; Table S3). For example, cladocerans were retained because they were highly correlated with only one other group (siphonophores) whereas siphonophores were dropped because they were highly correlated with six other groups (Table S3).”

While a different analysis, it would be interesting to know which of these taxa co-occur given certain conditions and which are responding to environmental variability differently. I suppose this can be inferred from the analysis herein, since the highly correlated taxa did in fact occur under the same conditions, but I guess I was thinking of a community analysis type approach.

This is a good question, and one which we plan to consider in the more detailed analysis of the within-year variability of zooplankton in the Strait of Georgia from 2015 to 2019, for which we have more samples. 

Line 223: patterns - corrected

Line 227: 1-9 combinations of factors? This makes it sound as if covariates were added here, but I think you mean fitting the model with 1-9 underlying trends? As in the abstract the term “factor” seems unusual here—the MARSS manual refers to “hidden trends” or “latent trends” with “factor loadings” estimated for the individual drivers of the trends (which observed TS is +/- for the latent trends).

Agreed. Replaced “factors” with “latent trends”

Line 251: N is Newton? Maybe I should know this, but don’t deal with wind much—maybe other more biologically-oriented people don’t either.

The reviewer is correct. Replaced “N” with “Newtons”

Line 305: Perhaps add “given their declines and failure to rebound” or something similar.

Agreed. We have added the following to that sentence: “…because of their strong declines in the 1990’s and failures to recover (e.g. [4])”.

Lines 352-388: This is a really nice description of the methods employed. A few points needing clarification: 

- line 354, describe the KPSS test a bit; We have added the following: “Regression statistics of the type applied in this study require the time series to be stationary around a mean or linear trend, i.e. that the mean and variance are constant over time.”

- line 369, define p,d,q; We have made this sentence more specific: “For example, for ARIMA order (p,d,q), “p” indicated whether autocorrelation was important (p>0), “d” indicated whether differencing of successive values was required (d>0), and “q” indicated whether a moving average term was needed (q>0).”

- and last line, is the more complex model the one with more covariates? Or something else? The reviewer is correct. We have changed this line to read “…variance (a significant ANOVA value meant the model with more covariates was the better fit).”

Results:

Line 438-442: This seems more appropriate for the discussion.

This paragraph is now redundant to the analyses presented in the new S1 file (see response to reviewer’s comments below regarding the uneven annual sampling). We have deleted this paragraph from the manuscript. 

Line 448-449: Is there no evidence of daytime net avoidance for other species? I realize the Euphs are more susceptible because of their size and diel migration, but is this just a case of the studies not having been done for other species or do they really not exhibit it?

This question is better addressed in the Taxonomic Analysis section of Methods. We now include the following: 

“Net avoidance by other taxa will depend on their size and swimming abilities. Calanoid copepods and amphipods are the more common larger zooplankton (along with euphausiids) in the Strait of Georgia. However, Shaw and Robinson [20] found few significant day-night differences in catchability of calanoid copepods by bongo nets in the study region, and Keen [21] concluded that net avoidance by copepods and amphipods on the north coast of British Columbia was a negligible concern for his study (although he used a plummet net with similar design to our SCOR net; plummet nets sample on the downcast). Consequently, we corrected for daytime net avoidance only for euphausiids.”

Line 510-ish: It is interesting to me that euphausiids did not load on either of the prevalent trends despite their dominance in the samples. This is perhaps worth noting in the text. It’s on the Fig. 7, but not explicitly mentioned.

We thank the reviewer for pointing this out. We have added the following sentence to the end of that paragraph: “Despite being a major contributor to the total zooplankton biomass, the pattern of annual anomalies of euphausiids did not load strongly on zooplankton trends 1 or 2 (Fig. 7A).”

Line 551: This is an interesting finding!

While the focus of this study was on the zooplankton, and their relationships with their physical environment and selected salmon marine survivals, we agree that there are interesting findings regarding the physical variables themselves. We have therefore added a sentence to the Conclusions section which emphasizes this point: “Two latent trends were also identified from the 10 physical variables retained in the analysis. Physical trend 1 represented a sea temperature and event timing (spring bloom peak, day of peak flow in the Fraser River) axis, and trend 2 represented a sea surface salinity and vertical stratification axis.”

Line 554: Please explain how figure 8A demonstrates 4 groups. I would think this figure demonstrates a preference for more groups. (Ah, I see it is in the caption—maybe move to text?)

We have changed this sentence to read: “The results of the constrained (chronological) cluster analysis based on the two latent zooplankton trends and the two latent physical trends revealed a maximum of four significant clusters (Fig. 8A; note that the largest number of significant clusters was determined as the highest number of groups whose sum of squares exceeded the values obtained from a randomisation process).”

Discussion:

Line 705-706: Consider referencing recent work by Litzow et al. discussing non-stationarity and PDO in the Gulf of Alaska and how local observations are related and tying your observations to what's happening there. (https://royalsocietypublishing.org/doi/full/10.1098/rspb.2018.1855, https://esajournals.onlinelibrary.wiley.com/doi/full/10.1002/ecy.2760#support-information-section)

We agree this is a useful paper to underline the problems with non-stationary data, and have cited it in the Discussion addressing the limitations of our study: “Therefore, if conditions change, the relationships derived from this study may lose their predictive skill (e.g. [62,63]).”

Line 737: We typically think of salmon as rather surface oriented, yet the zooplankton data are all collected from deeper stations. Are the zooplankton in and of themselves driving the salmon survival, or are they another indicator of some other process? Have the salmon surveys found outmigrants at the depths of the zooplankton you sampled in the Strait of Georgia? It seems that a comment on the depth relationship would help tie this together a bit better.

The reviewer is correct. We have added the following sentences to the Discussion section regarding the limitations of this study:

“In addition, it is worth noting that salmon generally are surface-oriented, with juveniles in coastal waters usually occurring between the surface and 15 m [60], and that they feed mostly at dawn or dusk [61]. In contrast, our zooplankton tows covered the full water column during both daylight and night. Such a tow pattern was necessary to produce a consistent series of zooplankton data over time, unaffected by variations in the depth range of the tow. As a consequence, however, not all the zooplankton that we sampled were necessarily available to the juvenile salmon, although with vertical migration of the zooplankton most will be in the upper water layers at dawn and dusk when juvenile salmon are actively feeding. We are unable, therefore, to definitively conclude whether the relationships identified in this study are a result of direct effects of zooplankton on salmon, or indirect effects due to correlations of both zooplankton and salmon with other variables.”

Line 790: I like this discussion of oceanography and production. Is there any evidence that the densities of zooplankton observed (especially the large guys) would be limiting for salmon?

This is a good question, but to address it properly is beyond the scope of this study. For example, using back-of-the-envelope calculations:

• juvenile Coho require about 3% of their body weight per day (Brodeur and Pearcy, 1987, J. Fish Biology 1987;31: 465-477)

• the average size weight of juvenile Coho in the Strait of Georgia in September 1997 to 1999 was about 166 g (Beamish and Mahnken, 2001, Progr in Oceanogr 41: 423-437)

• therefore the average prey biomass required per day by juvenile Coho in the Strait of Georgia is about 166 g * 0.03 � 5 g/day/Coho

• the Total zooplankton biomass from 1996 to 2018 was about 20 g/m2 (this study)

• the area of the Strait of Georgia is about 6800 km2

• therefore the biomass of All zooplankton in the Strait of Georgia is about 6.9*109 m2 * 20 g = 138*109 g

• the average abundance of juvenile Coho in the Strait of Georgia in September during 1997 to 1999 was 3,556,000 (Beamish and Mahnken, 2001)

• this number of juvenile Coho would require 3,556,000 * 5 g/day = 17,780,000 g of food per day (= 0.01778*109 g)

• therefore, on a daily basis, it would appear there was 138*109 g of total zooplankton available to meet a requirement of 0.01778*109 g of food for juvenile Coho. This amount of zooplankton prey would seem to be almost 10 times (138*109 / 0.01778*109 = 7.8) greater than that required for the abundance of juvenile Coho in the Strait of Georgia.

• However, this analysis does not consider that juvenile Coho eat prey other than zooplankton in the Strait of Georgia, nor that there are other animals in the Strait which will be feeding on the zooplankton.

We therefore consider that addressing this question involves many other data sources and assumptions, and to do it correctly is outside the scope of this paper. 

Line 832: There is rather extreme uneven sampling effort in this time series and while I think good effort has been made to neutralize differences, I’m still wondering if this has bearing on the outcomes of the analysis. It would be interesting to look at the most recent years where sampling has been most robust and address variation within the samples. Surely additional samples increase the variation as the authors point out, but also they allow inference in different ways and at different scales. Perhaps other work is being done on these samples (the “further study”) that would allow for understanding community change over short time scales—this may in turn yield insight on a longer time scale. It would be nice if the authors could comment on this and whether the small sample sizes early on are “good enough” for drawing conclusions, or whether the high-intensity sampling has/is expected to contributed additional conclusions.

This is an excellent question, which encouraged us to run a numerical simulation of the potential impact of the uneven number of samples each year. This simulation is described in S1 File. We have added the following to the Results - Zooplankton section: 

“It is possible that the uneven sampling effort among years may have influenced the interannual pattern of total zooplankton biomass anomalies. There are two questions: (1) Could the larger number of samples from 2015 to 2018 (average of 112 samples per year) have influenced the biomass anomalies compared, for example, with the period from 2003 to 2007 when only 12 samples on average were collected each year? (2) What is the probability that the sequence of low total zooplankton biomass anomalies from 2003 to 2007 (5 years in a row with anomalies less than -0.10, Fig. 2D) would have occurred if the true distribution of samples was similar to that in 2015 to 2018 (i.e. with true annual anomalies near or somewhat above the climatological mean, 1996-2010) and 12 samples had been collected each year? These questions were assessed using a numerical simulation with 1000 repetitions (S1 File). The simulations demonstrate that, for question 1, the anomalies calculated using the full data set for 2015 to 2018 were almost identical to the mean anomalies that would likely have been observed if only 12 samples had been collected each year. It is therefore unlikely that the higher anomalies in 2015 to 2018 resulted only from the larger number of samples in those years, compared with the number of samples collected from 2003 to 2007. For question 2, the probability of getting five biomass anomalies in a row with values less than -0.10, as was observed from 2003 to 2007 (Fig. 2D), if the true distribution of total zooplankton biomass was as observed in 2015 to 2018, and based on 12 samples per year, is highly unlikely (P=0.00032; S1 File). We conclude that the total zooplankton biomass, and therefore the biomass anomalies, in 2003 to 2007 was lower than the biomass observed in 2015 to 2018, even though an average of only 12 samples were collected each year.”

Line 841: This is a thorough summary but also an extremely long paragraph and could be broken up.

We have now titled this section as “Conclusions” and separated it into three paragraphs.

Line 871: This suggestion brings me back to the varying levels of effort and if there is any way to leverage that variation in determining what such a monitoring program would look like…just food for thought (if the team needs something to do).

This is a great suggestion, and one which we will plan to consider in a further study looking in greater detail at the seasonality of the zooplankton using data from 2015 to 2019. 

Figs.

In figure 3, panels E and F should be pulled out and put in their own 2-panel plot so the reader has some hope of actually seeing the composition.

This has been addressed by removing panels A to D from Fig. 3 in the main text and including them in the new S1 File addressing the uneven annual sampling issue (see response to reviewer’s comment, above). 

In addition, we have made minor corrections and changes throughout the text to improve readability. 

We look forward to your, and the reviewers’, comments on this revised manuscript. 

Sincerely yours,

Ian Perry

---

## [Decision Letter · Decision Letter 1]

11 Jan 2021

Zooplankton variability in the Strait of Georgia, Canada, and relationships with the marine survivals of Chinook and Coho salmon

PONE-D-20-31572R1

Dear Dr. Perry,

We’re pleased to inform you that your manuscript has been judged scientifically suitable for publication and will be formally accepted for publication once it meets all outstanding technical requirements.

Kind regards,

Syuhei Ban

Academic Editor

PLOS ONE

Additional Editor Comments (optional):

Reviewers' comments:

Reviewer's Responses to Questions

**Comments to the Author**

1. If the authors have adequately addressed your comments raised in a previous round of review and you feel that this manuscript is now acceptable for publication, you may indicate that here to bypass the “Comments to the Author” section, enter your conflict of interest statement in the “Confidential to Editor” section, and submit your "Accept" recommendation.

Reviewer #1: All comments have been addressed

Reviewer #2: All comments have been addressed

2. Is the manuscript technically sound, and do the data support the conclusions?

Reviewer #1: Yes

Reviewer #2: Yes

3. Has the statistical analysis been performed appropriately and rigorously? 

Reviewer #1: Yes

Reviewer #2: Yes

4. Have the authors made all data underlying the findings in their manuscript fully available?

Reviewer #1: Yes

Reviewer #2: Yes

5. Is the manuscript presented in an intelligible fashion and written in standard English?

Reviewer #1: Yes

Reviewer #2: Yes

6. Review Comments to the Author

Reviewer #1: (No Response)

Reviewer #2: I am pleased with the corrections and think this paper will make a solid contribution to marine science in the region.

I especially appreciate:

1. The response related to zooplankton availability—while the back of the envelope exercise may not be publication-worthy, I appreciate the effort and would support further investigation into this idea.

2. The simulation experiment regarding sample size. The effort strengthens the results presented in the paper.

7. PLOS authors have the option to publish the peer review history of their article (what does this mean?). If published, this will include your full peer review and any attached files.

Reviewer #1: No

Reviewer #2: No

---

## [Editor Report · Acceptance letter]

15 Jan 2021

PONE-D-20-31572R1 

Zooplankton variability in the Strait of Georgia, Canada, andrelationships with the marine survivals of Chinook and Coho salmon 

Dear Dr. Perry:

I'm pleased to inform you that your manuscript has been deemed suitable for publication in PLOS ONE. Congratulations! Your manuscript is now with our production department. 

Kind regards, 

on behalf of

Dr Syuhei Ban 

Academic Editor

PLOS ONE